# In-Context Occam's Razor: How Transformers Prefer Simpler Hypotheses on the Fly

**Puneesh Deora**[1], **Bhavya Vasudeva**[2], **Tina Behnia**[1], **Christos Thrampoulidis**[1]
[1] University of British Columbia, Vancouver
[2] University of Southern California, Los Angeles
{puneeshdeora,tina.behnia,cthrampo}@ece.ubc.ca, bvasudev@usc.edu

## Abstract

In-context learning (ICL) enables transformers to adapt to new tasks through contextual examples without parameter updates. While existing research has typically studied ICL in fixed-complexity setups, real-world language models encounter tasks of diverse complexity levels. This paper investigates how transformers navigate hierarchical task structures where higher-complexity categories can perfectly represent any pattern generated by simpler ones. We design testbeds based on Markov chains and linear regression that reveal transformers not only identify the correct complexity level for each task but also accurately infer the corresponding parameters—even when the in-context examples fit multiple complexity hypotheses. Notably, when presented with data generated by simpler processes, transformers consistently favor the least complex sufficient explanation. We theoretically explain this behavior through a Bayesian framework, demonstrating that transformers effectively implement an in-context Bayesian Occam's razor by balancing model fit against complexity penalties.

## 1 Introduction

In-context learning (ICL) has emerged as a fundamental capability of large language models (LLMs), enabling them to adapt to novel tasks through contextual examples without parameter updates (Brown et al., 2020). This remarkable ability has drawn significant attention in interpretability research, with studies examining both commercial-scale LLMs (Elhage et al., 2021; Wang et al., 2023; Min et al., 2022) and controlled, synthetic environments. The latter approach trains transformers from scratch on well-defined tasks—such as linear regression (Garg et al., 2022; Akyürek et al., 2023; von Oswald et al., 2022; Zhang et al., 2023), discrete functions (Bhattamishra et al., 2024), and Markov processes (Edelman et al., 2024; Rajaraman et al., 2024; Park et al., 2025)—offering precise control over training distributions and enabling direct comparisons with known algorithms. However, these synthetic studies typically restrict analysis to tasks of *fixed complexity*, diverging from real-world scenarios where LLMs encounter diverse tasks spanning multiple complexity levels.

To bridge this gap, we investigate ICL in setups with hierarchical task complexity structures. We design task categories with *distinct complexity* levels, where higher-complexity categories form strict supersets of lower ones: specifically, the higher-complexity category contains hypotheses capable of emulating any pattern produced by simpler categories. For instance, consider a transformer trained on next-token prediction for sequences generated by both order-1 and order-3 Markov chains. Since any order-1 chain can be represented as a special case of an order-3 chain, an inherent ambiguity arises during inference: when presented with a sequence genuinely generated by an order-1 process, can the transformer identify the true complexity class, or will it default to the most expressive hypothesis in its repertoire?

*Can transformers effectively differentiate between tasks of varying complexity during in-context learning? When presented with data compatible with multiple hypothesis classes, do they accurately identify the simplest sufficient hypothesis, or do they systematically default to the most complex available hypothesis?*

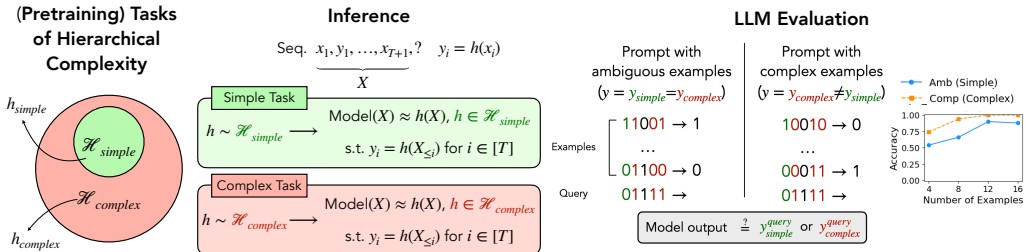

Figure 1: **(Left)** We train transformers on a hierarchy of task families in which the complex class strictly contains the simpler one (the simple category). **(Middle)** After training, the models use the in-context examples to infer both the appropriate complexity level and the task parameters that best fit the context, on the fly. **(Right)** We probe the same inductive bias in off-the-shelf LLMs: prompted with ambiguous examples that admit both a simple and a complex explanation (copy the first bit or take majority over three bits (second, fourth and fifth), respectively), the model performs in-context Occam's razor by consistently following the simple hypothesis (see Section 3.4 for details.).

Our systematic investigation answers this question affirmatively. See Fig. 1 for an overview of our framework. Returning to our Markov chain example, Fig. 2 demonstrates that at inference time, the transformer successfully recognizes the true order of the generating process. When presented with a sequence from an order-1 chain, (left) it appropriately employs bigram statistics; when presented with an order-3 chain, (right) it switches to tetragram statistics. This confirms that transformers can distinguish between different complexity categories and adapt their predictions to the context, rather than defaulting to the highest-complexity hypothesis available.

Our contributions are as follows:

- We introduce a framework for studying ICL across hierarchical complexity levels, where higher-complexity tasks form strict supersets of simpler ones, creating inherent ambiguity in hypothesis selection in context.
- We empirically demonstrate in two representative testbeds, a Markov chain and a linear regression setting, that transformers correctly identify the simplest sufficient hypothesis rather than defaulting to the more expressive one.
- We provide a theoretical justification through a Bayesian lens, showing that transformers naturally implement Bayesian Occam's razor in-context by balancing data likelihood against model complexity.
- Additionally, going beyond the two testbeds, we demonstrate the same behaviour on transformers trained on probabilistic context-free grammars (PCFGs) with varying complexity, and on pretrained LLMs (GPT-4) tested on a representative setting with Boolean-function tasks.

## 2 Background and Previous Work

To understand the mechanisms underlying ICL, researchers have developed controlled synthetic environments with well-defined tasks. In just the last couple of years, there has been a proliferation of work on this topic. Here, we focus on reviewing two key synthetic setups: Markov chains and linear regression, introduced by Edelman et al. (2024) and Garg et al. (2022), respectively, which form the basis for our investigation into tasks of varying complexity. We then discuss follow-up studies most relevant to our analysis. For a comprehensive review and additional references, we refer readers to Dong et al. (2024).

### 2.1 Markov Chains and Linear Regression as Testbeds for ICL

**ICL of Markov Chains.** Markov chains were introduced as a testbed for studying ICL by Edelman et al. (2024). Specifically, they showed that transformers trained on sequences generated from random order-1 Markov chains learn to do in-context inference on unseen Markov chains of order-1 by learning to output bigram statistics of the context.

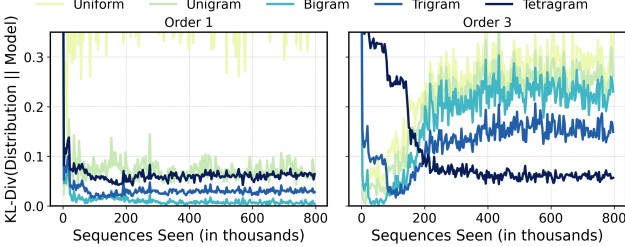

Figure 2: We train a transformer for next-token prediction on sequences generated by random order-1 and order-3 Markov chains. During inference, we assess its ICL ability by evaluating performance on sequences derived from unseen order-1 and order-3 chains. **(Left)** shows the distance between the model's output distribution on the last token and $n$-gram statistics of the context for order-1 inference, as a function of the number of sequences seen during training. **(Right)** presents analogous results for order-3 inference. *Notably, the trained transformer can identify the true order (1 or 3) of the context, and then predict using either bigram or tetragram statistics accordingly.*

To establish the foundation for our study, we now formalize their key findings. Let $x_t$ denote the $t$-th symbol from a finite vocabulary $[V] = \{1, ..., V\}$ of size $V$ generated by a $k$-th order Markov chain, *i.e.*, $p(x_t = v|x_{t-1}, ..., x_1) = p(x_t = v|x_{t-1}, ..., x_{t-k+1})$ for all $v \in [V]$ and let $P \in \mathbb{R}^{V^k \times V}$ denote the row-stochastic transition matrix with these conditional distributions as rows. Sample a transition matrix at random: for example, sample each of its rows according to a Dirichlet prior with parameter $\boldsymbol{\alpha}$. Generate a sequence $X := X_{\leq T} = [x_1, x_2, ..., x_{T-1}]$ of $T \gg k$ symbols from this transition matrix by letting the first $k$ entries $\{x_1, ..., x_k\}$ be drawn uniformly at random. A transformer model $M_\theta$ parameterized by $\theta$ is trained to auto-regressively predict the next-symbol $x_{t+1}$ using the previous $t$ symbols $X_{\leq t}$ by (approximately) minimizing the expected loss, with respect to cross-entropy $\ell$:

$$L(\theta) := \mathbb{E}_{X \sim P \sim \text{Dir}(\mathbf{1})^{\otimes V}} \left[ \sum_{t=1}^{T-1} \ell(M_\theta(X_{\leq t}), x_{t+1}) \right], \tag{1}$$

where $\text{Dir}(\mathbf{1})^{\otimes V}$ denotes $V$ independent draws from the Dirichlet prior. At inference, draw transition matrix $P_{\text{inf}} \sim \text{Dir}(\mathbf{1})^{\otimes V}$, and let $X \sim P_{\text{inf}}$ denote a sequence of length $T - 1$ generated from it that is given as prompt to the model. Edelman et al. (2024) show that the Kullback–Leibler (KL) divergence of the model's output probability to the birgram statistics

$$\sum_{t=2}^{T-1} \mathbb{1}(x_{t-1} = x_T, x_t = v) \Big/ \sum_{t=2}^{T} \mathbb{1}(x_{t-1} = x_T), \quad v \in [V], \tag{2}$$

averaged over many realizations of $X, P_{\text{inf}}$, is (almost) zero. Here, $\mathbb{1}(\cdot)$ is the indicator function that equals 1 when the condition is true and 0 otherwise. Thus, the transformer looks back at the sequence to compute the bigram probabilities, and predict the next token $v \in [V]$ using these probabilities.

**ICL of Linear Regression.** Before Markov chains, the first synthetic playground for ICL was introduced by Garg et al. (2022), who demonstrated that a transformer trained on random examples from linear regression tasks can learn to perform in-context inference on unseen tasks by computing the least-squares solution. In this setup, sequences consist of interleaved input-output pairs $(\mathbf{x}, y)$ of the form $X = [\mathbf{x}_1, y_1, \mathbf{x}_2, y_2, ..., \mathbf{x}_t, y_t]$[1], with feature vectors $\mathbf{x}_t \in \mathbb{R}^d$ and labels $y_t \in \mathbb{R}$. The feature vectors $\mathbf{x}_t$ are sampled i.i.d from a standard Gaussian distribution, *i.e.*, $\mathbf{x}_t \sim \mathcal{N}(\mathbf{0}, \mathbb{I}_d)$. Each sequence is characterized by a task-specific weight vector $\mathbf{w} \sim \mathcal{N}(\mathbf{0}, \mathbb{I}_d)$, such that labels are generated as $y_t = \mathbf{w}^\top \mathbf{x}_t, \forall t \in [T]$.

Similar to the Markov chain setup, the transformer $M_\theta$ is trained to predict label $y_{t+1}$ using the previous $t$ pairs $(\mathbf{x}, y)$ along with the new input $\mathbf{x}_{t+1}$ (collectively denoted by $X_{\leq t}$), by

---

[1]To be precise, the model input is $[\mathbf{x}_1, \mathbf{y}_1, ..., \mathbf{x}_t, \mathbf{y}_t]$, where $\mathbf{y}_i \in \mathbb{R}^d$, with zeros in the first $d - 1$ entries and $y_i$ as the last entry.

minimizing the expected loss over sequences with $T$ pairs:

$$L(\theta) := \mathbb{E}_{\mathbf{x}_t \sim \mathcal{N}(\mathbf{0}, \mathbb{I}_d), \, \mathbf{w} \sim \mathcal{N}(\mathbf{0}, \mathbb{I}_d)} \left[ \sum_{t=1}^{T-1} \ell(M_\theta(X_{\leq t}), y_{t+1}) \right], \tag{3}$$

where $\ell$ is the squared loss. Garg et al. (2022) demonstrated that when presented with an inference-time sequence $X = [\mathbf{x}_1, y_1, \mathbf{x}_2, y_2, ..., \mathbf{x}_{T-1}, y_{T-1}, \mathbf{x}_T]$ generated from an unseen task vector $\mathbf{w}_{\mathrm{inf}} \sim \mathcal{N}(\mathbf{0}, \mathbb{I}_d)$, the transformer effectively infers $\mathbf{w}_{\mathrm{inf}}$ from the in-context examples. Specifically, the squared error between the model's prediction and the least-squares prediction $\hat{y}_{\mathrm{LS}} = \mathbf{x}_T^\top \mathbf{w}_{\mathrm{LS}}$ where $\mathbf{w}_{\mathrm{LS}} = A^\dagger \mathbf{y}$ approaches zero when averaged over many realizations of $X$ and $\mathbf{w}_{\mathrm{inf}}$. Here, $A \in \mathbb{R}^{(T-1) \times d}$ represents the feature matrix with columns $[\mathbf{x}_1, \mathbf{x}_2, \ldots, \mathbf{x}_{T-1}]^\top$, $\mathbf{y} \in \mathbb{R}^{T-1}$ is the vector of context labels $[y_1, \ldots, y_{T-1}]$, and $A^\dagger$ denotes the Moore-Penrose pseudoinverse.

**Other Related Work.** Raventos et al. (2023) and Park et al. (2025) study linear-regression and Markov-chain settings, respectively, training transformers on sequences drawn from a finite task set and show that limited task diversity induces a trade-off between task retrieval and task learning modes of ICL (Pan et al., 2023). On the theoretical side, Xie et al. (2022) give a Bayesian account of ICL, and a complementary line of work provides explicit transformer weight constructions that implement functionalities observed to facilitate ICL—covering linear regression (Li et al., 2024; 2023; Ahn et al., 2023; von Oswald et al., 2022; Fu et al., 2024) and Markov chains (Edelman et al., 2024; Rajaraman et al., 2024; Chen et al., 2024b). All of these studies focus on a single task family; none tackle settings where multiple task categories are linked by a hierarchy of complexity. We bridge this gap by extending the Bayesian lens to hierarchically related tasks and showing that transformers consistently adopt the simplest hypothesis that explains the context in both regression and Markov-chain domains. Transformers have also been shown to perform algorithm selection: Lin & Lee (2024) reveal retrieval-versus-learning modes in Gaussian-mixture regression, and Bai et al. (2023) show context-dependent switching among linear, noisy-linear, and classification rules. A broader simplicity view links ICL to prequential coding (Elmoznino et al., 2025), while Xiong et al. (2024) study prompts containing several unrelated tasks in superposition. However, none of these studies tackle hierarchically related task categories, where a higher-complexity class can fully explain the context produced by any simpler class. We consider this challenging regime and show that transformers resolve the ambiguity through a Bayesian Occam's razor, consistently defaulting to the simplest hypothesis that still explains the context. See Appendix E for a detailed discussion.

## 3 In-Context Learning Across Hierarchical Complexity Levels

Previous research has extensively examined ICL for fixed complexity tasks. But how do transformers behave when trained on tasks spanning multiple well-defined complexity levels? Can they accurately identify both the complexity category of a given task and infer its underlying parameters in-context? Here, we introduce two testbeds—extensions of the Markov chain and linear regression paradigms—to systematically investigate these questions. Additionally, we verify the conclusions on a probabilistic grammar setting.

### 3.1 Markov Chains

**Modeling Tasks of Varying Complexity.** To form task categories of varying complexity, we train on Markov chains of different orders, where we group all chains of a fixed order into one task category. The category of order-$k_1$ has lower complexity than any order-$k_2$ category for $k_2 > k_1$ due to fewer degrees of freedom. Importantly, any order-$k_1$ chain can be perfectly represented as a special case of an order-$k_2$ chain, making higher-order categories strict supersets of lower-order ones.

For concreteness, we train the transformer model $M_\theta$ on sequences generated from two different task categories[2]. We consider order-1 and order-$k$ categories, where $k > 1$. During training, we first sample a transition matrix $P_s$ of order-$s$, which is then used to generate a

---

[2]Multiple task categories can be treated in a similar manner without substantial further insights. Thus, we focus on two categories for simplicity of exposition.

sequence $X = [x_1, x_2, ..., x_T]$ of length $T$. The order $s$ is sampled uniformly at random from the set ord $= \{1, k\}$. Similar to Eq. (1), we train autoregressively to minimize the (expected) loss, this time over Markov chains of both orders $\{1, k\}$, again with cross-entropy loss $\ell$, i.e.,

$$L(\theta) := \mathbb{E}_{\substack{X \sim P_s \sim \text{Dir}(\mathbf{1})^{\otimes V^s} \\ s \sim \text{Unif}(\text{ord})}} \left[ \sum_{t=1}^T \ell(M_\theta(X_{\leq t}), x_{t+1}) \right]. \tag{4}$$

At inference, we prompt the model with a sequence generated from either order-1 (simple) or order-k (complex) chain. To evaluate the ICL ability of the model, we compare (with respect to KL divergence) the model's output probability for the token following the prompt to either the order-1 statistics of Eq. (2) or the order-k statistics

$$\sum_{t=k}^{T-1} \frac{\mathbb{1}(x_{t-1} = x_T, x_{t-2} = x_{T-1}, ..., x_{t-k+1} = x_{T-k}, x_t = v)}{\sum_{t=k}^T \mathbb{1}(x_{t-1} = x_T, x_{t-2} = x_{T-1}, ..., x_{t-k+1} = x_{T-k})}, \ v \in [V]. \tag{5}$$

When prompted with an order-$k$ chain, the transformer must recognize the higher-order dependencies that cannot be captured by order-1 statistics. However, when prompted with an order-1 chain, the transformer faces a more subtle decision, as any order-1 transition matrix can be perfectly represented as a special case of an order-$k$ transition matrix. This latter scenario directly tests whether the transformer defaults to the most complex hypothesis or correctly identifies the simplest sufficient one.

**Transformers Distinguish Between Task Categories In-context.** We train a GPT-2 type decoder-only transformer for all the experiments (Karpathy, 2023). Please refer to Appendix C for specific training details.

In Fig. 2, we train a transformer on sequences drawn from order-1 and order-3 Markov chains (i.e., Eq. (4) with $s = \{1, 3\}$). We then measure the KL divergence between the transformer's output distribution and several well-defined statistical strategies when the input is taken from unseen order-1 or order-3 Markov chains: uniform, bigram, trigram, tetragram statistics (Eq. (5) for $k = 0, \ldots, 3$). We see clearly that with sufficient training, the transformer consistently learns to distinguish between order-1 and order-3 sequences, applying bigram and tetragram statistics appropriately to each case.

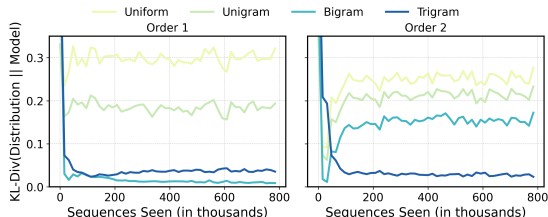

Figure 3: A transformer trained on sequences generated by random order-1 and order-2 Markov chains can infer whether the context is order-1 or order-2, then generate predictions using the corresponding bigram or trigram statistics. **(Left)** shows the distance between the model's output distribution on the last token and well-defined context strategies for order-1 inference, as a function of the number of sequences seen during training. **(Right)** presents analogous results for order-2 inference.

We validate this finding in Fig. 3 by repeating the experiment with order-1 and order-2 Markov chains, again observing that the model accurately infers the underlying order. Crucially, in both experiments, the transformer does not default to the highest-order statistics (tetragram or trigram) for all inputs, despite these statistics having sufficient expressive power to model order-1 sequences. Instead, it selects the appropriate complexity level based on the in-context sequence. This demonstrates that the transformer effectively implements Occam's razor, identifying the simplest hypothesis adequately explaining prompt.

In these experiments, the context length $T$ is kept fixed at $T = 300$ (Fig. 2) and $T = 200$ (Fig. 3) for both training and inference. Here, we focus on how the transformer learns to distinguish between the two categories as a function of training iterations (equivalently, sequences processed). In Section 4, we provide additional experiments examining how performance varies with context length.

**Bayesian Interpretation of the Selection Mechanism.** We adapt a Bayesian viewpoint of ICL (Panwar et al., 2024; Xie et al., 2022; Lin & Lee, 2024) to explain how transformers

choose the simplest task that explains the context. In this respect, we assume sufficient training data such that the transformer implements the Bayes optimal minimizer. Recall the mixture model described in Eq. (4), and let $X = [x_1, \ldots, x_T]$ be an observed sequence for $T \gg k$. Then, the *Bayes' optimal* predictive distribution for $x_{T+1}$ given $X$ is the mixture

$$\underbrace{p(x_{T+1} = v \mid X)}_{\text{model output}} = \sum_s \underbrace{p(s \mid X)}_{\text{category posterior}} \cdot \underbrace{p(x_{T+1} = v \mid X, s)}_{s\text{-gram statistics of } X}, \quad v \in [V], \tag{6}$$

where $p(s \mid X)$ denotes the posterior probability of the order-$s$ chain given the sequence. Further, $p(x_{T+1} = v \mid X, s)$ denotes the Bayes' optimal predictive distribution computed with respect to only order-$s$ Markov chains. Specifically, for the Dirichlet prior $\text{Dir}(\mathbf{1})^{\otimes V^s}$ it can be shown that $p(x_{T+1} = v \mid X, s)$ which is a smoothed version of the order-$s$ statistics in Eq. (5); e.g. see Edelman et al. (2024, Eq. (3)). When $T \gg k$, where both statistics are well-defined for both orders 1 and $k$, the smoothing effect becomes negligible (particularly since the smoothing is essentially independent of $s$).

From Eq. (6), the model's output (LHS) when trained with chains of all orders $s$ can be seen as a *convex mixture* of what would have been the model's output if the transformer was trained only on a single order. This means the model's output crucially depends on the posterior probability coefficients $p(s \mid X)$, which we can express as: $p(s \mid X) = p(X \mid s) \pi(s) / \sum_{s' \in \text{ord}} p(X \mid s') \pi(s')$. Here, $\pi(s)$ is the prior on orders, which is set to be uniform in Eq. (4) and thus cancels out in the ratio, and $p(X \mid s)$ is the marginal likelihood of the prompt (context) given the order $s$. Consequently, which category's statistics ($s$-gram) the transformer's outputs depends on which marginal likelihood $p(X \mid s)$ dominates.

*Asymptotics of Likelihood.* For large $T \gg k$, standard Dirichlet–multinomial conjugacy combined with a Bayesian Information Criteria (BIC) style Laplace approximation (Schwarz, 1978; Ghosal & van der Vaart, 2017; Csiszár & Talata, 2006) gives the following useful approximation to the marginal likelihood (Schwarz, 1978):

$$\log p(X \mid s) \approx \sum_t \log \hat{p}_X(x_t \mid x_{t-1}, \ldots, x_{t-s}) - \frac{V^s(V-1)}{2} \log T. \tag{7}$$

Here, $\hat{p}_X$ denotes the *empirical* conditional probabilities derived from the prompt $X$. See Appendix B.1 for details. Thus, the marginal likelihood decomposes into an empirical likelihood term and a model complexity penalty term.

*Bayesian Occam's Razor.* Suppose the data is generated by an order-$s^*$ Markov chain, representing a simple category. We wish to compute the ratio $p(X \mid s)/p(X \mid s^*)$ for any complex category of order-$s$ with $s > s^*$. Note that for $T \gg k$, for any $s \geq s^*$ the empirical probabilities $\hat{p}_X(x_t \mid x_{t-1}, \ldots, x_{t-s})$ approach the true data transition probabilities derived from the simple category $s^*$. Thus, the empirical likelihood terms in Eqn. (7) are (approximately) equal for any $s \geq s^*$. This reflects the fundamental property that a Markov chain of order $s^*$ can be perfectly represented as a Markov chain of order $s$ where the transition probabilities are invariant to the additional history. However, the model complexity penalty term being proportional to $V^s$ strongly favors the smaller $s$. Together, for $T \gg k$ it holds:

$$\forall s > s^* : \frac{p(X \mid s)}{p(X \mid s^*)} \approx 0 \implies p(s^* \mid X) \approx 1 \overset{Eqn.(6)}{\implies} \underbrace{p(x_{T+1} = \cdot \mid X \sim P_{s^*})}_{\substack{\text{model output conditioned on} \\ \text{prompt from simple category}}} \approx \overset{s^*\text{-gram}}{\underset{\text{statistics}}{}}. \tag{8}$$

The model implements *Bayesian Occam's razor* in favor of the true lower-order model $s^*$.

*Saturation to the Higher Order.* Now, suppose that the true process is of order-$k$, rather than (say) order-1, where $k > 1$. The model order complexity term in Eq. (7) always favors the lower order (here $= 1$). However, in this case the empirical likelihood terms differ to each other when evaluated for $s = k$ vs $s = 1$. Previous works (Ghosal & van der Vaart, 2017; Csiszár & Talata, 2006) have shown that the *per-sample* improvement in log likelihood under the correct, higher-order model accumulates linearly in $T$. Concretely, for large $T$ and some universal constant $c > 0$,

$$\log p(X \mid s = k) - \log p(X \mid s = 1) \approx c\,T - \frac{(V^{k-1}-1)V(V-1)}{2} \log T \overset{T \gg k}{\implies} p(s = k \mid X) \approx 1.$$

Thus, the posterior probability correctly *saturates* to the true higher order once $T$ is large. In Appendix D, we construct a 2-layer transformer that can compute the empirical conditional probabilities $\hat{p}_X$, which are key to compute the posterior $p(s \mid X)$ needed to realize the Bayes' optimal solution.

## 3.2 Linear Regression

**Modeling Tasks of Varying Complexity**  We construct a similar setup of two task categories with varying complexity for ICL of linear regression. We define two categories with hierarchical complexity: a "complex" category parameterized by $\mathbf{w} = \mathbf{w}_d \sim \mathcal{N}(\mathbf{0}, \mathbb{I}_d)$ using the full feature space, and a "simple" category where $\mathbf{w}$ lies on a lower-dimensional subspace of $\mathbb{R}^d$ with the structure $\mathbf{w} = [\mathbf{w}_{d/2}, \mathbf{0}]$, where $\mathbf{w}_{d/2} \sim \mathcal{N}(\mathbf{0}, \mathbb{I}_{d/2})$[3]. This simple category corresponds to a sparse linear regression where half the features have no impact on the output. The tasks in the complex category have higher degrees of freedom ($d$ vs. $d/2$) and thus greater expressive power. Importantly, the complex category forms a strict superset of the simple category in the sense that any sequence $X$ generated from the simple category can be perfectly explained by some parameter from the complex category. The training details are analogous to Eq. (3) only now we first sample $s$ uniformly from the set $\dim = \{d/2, d\}$, then generate $\mathbf{w}_s \sim \mathcal{N}(\mathbf{0}, \mathbb{I}_s)$ and generate the labels $y_t = \mathbf{w}^\top \mathbf{x}_t$ for $t \in [T]$ where for $s = d/2$, we effectively use $\mathbf{w} = [\mathbf{w}_s, \mathbf{0}]$ (zero-padded to dimension $d$). We refer the reader to Appendix B.2 for details.

At inference time, we generate prompts using either $\mathbf{w}_d$ or $\mathbf{w}_{d/2}$ as the underlying task parameters. Let $X$ denote a sequence with $T$ in-context examples $(\mathbf{x}_t, y_t)$, and $\mathbf{x}_{\text{test}}$ the query vector for which we want to predict a label. We compare the transformer's output to two least-squares (LS) benchmark estimates: $\hat{y} = \mathbf{x}_{\text{test}}^\top \mathbf{w}^{\text{LS}}$, where $\mathbf{w}^{\text{LS}}$ is either the full $d$-dimensional solution $\mathbf{w}_d^{\text{LS}} := A_d^\dagger \mathbf{y}$ or the restricted $d/2$-dimensional solution $\mathbf{w}_{d/2}^{\text{LS}} = A_{d/2}^\dagger \mathbf{y}$. Here, $A_d := [\mathbf{x}_1, \mathbf{x}_2, \ldots, \mathbf{x}_T]^\top$ is the feature matrix, $\mathbf{y} = [y_1, \ldots, y_T]$ is the label vector, and $A_{d/2} := A_d [\mathbb{I}_{d/2} \quad \mathbf{0}_{d/2}]^\top$ is the projection of the feature matrix onto the first $d/2$ dimensions. When context length $T < d$ or $T < d/2$ for the respective estimates, the LS solution corresponds to the minimum $\ell_2$-norm interpolating solution.

When prompted with a sequence generated by the complex regressor $\mathbf{w}_d$, the full-dimensional solution $\mathbf{w}_d^{\text{LS}}$ naturally provides a better fit than $\mathbf{w}_{d/2}^{\text{LS}}$, which is constrained to a $d/2$-dimensional subspace. However, the critical question arises when we prompt the model with a sequence generated by the simple regressor $\mathbf{w}_{d/2}$: Does the transformer default to the more expressive full-dimensional solution, or does it correctly identify the simpler hypothesis? Specifically when $d > T \geq d/2$, both solutions perfectly interpolate the context data, yet they generally differ in their predictions. This creates an ideal test for whether transformers implement a form of Occam's razor when selecting in-context between equally valid explanations of varying complexity.

**Transformers Distinguish Between Task Categories In-context**  Fig. 4 demonstrates how transformers distinguish between task complexities in linear regression. When presented with sequences generated by the complex regressor $\mathbf{w}_d$, the transformer's predictions consistently align with the full-dimensional solution $\mathbf{w}_d^{\text{LS}}$ rather than $\mathbf{w}_{d/2}^{\text{LS}}$, as expected since the restricted solution lacks sufficient degrees of freedom. The most interesting finding emerges when the transformer is prompted with sequences from the simple regressor $\mathbf{w}_{d/2}$ in the regime where $d > T \geq d/2$. Here, despite both $\mathbf{w}_d^{\text{LS}}$ and $\mathbf{w}_{d/2}^{\text{LS}}$ perfectly interpolating the context data, the transformer's predictions significantly favor $\mathbf{w}_{d/2}^{\text{LS}}$. This demonstrates that the transformer does not default to the more expressive full-dimensional solution but rather correctly identifies the simpler hypothesis—a clear demonstration of Occam's razor at work. As context length increases into the $T \geq d$ regime, we observe $\mathbf{w}_d^{\text{LS}} = \mathbf{w}_{d/2}^{\text{LS}} = \mathbf{w}_{d/2}$, and the transformer essentially recovers the true regressor.

---

[3]The specific choice of $d/2$ as the dimensionality and the alignment of the subspace with the coordinate axes are made for simplicity, without affecting the generality of our findings.

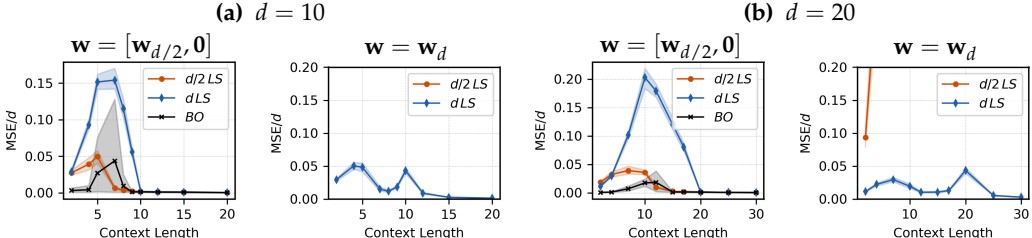

Figure 4: A transformer trained on $T = 39$-long sequences generated using random **(a)** $d = 10$ or **(b)** $d = 20$ dimensional regressors. We plot $(M_\theta([X, \mathbf{x}_{\text{test}}]) - \mathbf{w}^\top \mathbf{x}_{\text{test}})^2/d$ where $\mathbf{w}$ refers to the two benchmark least-squares solutions $\mathbf{w}^{\text{LS}}_{d/2}$ or $\mathbf{w}^{\text{LS}}_d$ in Section 3. When prompted with unseen sequences of **(left)** $d/2$ or **(right)** $d$ dimensional regressors, the predictions align most closely with the corresponding LS solution $\mathbf{w}^{\text{LS}}_{d/2}$ or $\mathbf{w}^{\text{LS}}_d$. In the right figures, (orange, dot) is off-scale, demonstrating that $\mathbf{w}^{\text{LS}}_{d/2}$ is a bad fit for sequences generated using $\mathbf{w}_d$. See  for evaluation over train time.

### 3.3 Probabilistic Grammars

In this section, we validate Occam's razor like inductive bias in transformers trained on probabilistic context-free grammars (PCFGs), which are stochastic systems designed to generate sequences of symbols from specified vocabularies (Collins, 2013) (see Appendix B.4 for a formal description). In our setup, non-terminal symbols $\mathsf{N} := \{\mathsf{S}, \mathsf{A}, \mathsf{B}, \mathsf{C}\}$, terminal symbols $\mathsf{T} := \{\mathsf{a}, \mathsf{b}, \mathsf{eos}\}$, where eos corresponds to the end-of-string symbol. The 'complex' PCFGs use the following set of probabilistic productions:

$$\mathsf{S} \to \mathsf{ABC} \mid \mathsf{BAC} \mid \mathsf{AAC} \mid \mathsf{BBC} \qquad \text{with probabilities } p_1, p_2, p_3, p_4,$$
$$\mathsf{A} \to \mathsf{a}, \quad \mathsf{B} \to \mathsf{b} \tag{9}$$
$$\mathsf{C} \to \mathsf{S} \mid \mathsf{eos} \qquad \text{with probabilities } q_1, q_2.$$

The 'simple' PCFGs are a subset of the 'complex' PCFGs, where we enforce $p_3 = p_4 = 0$.

To generate sequences from this set of PCFGs, we first draw the probabilities $p_i$ and $q_i$ randomly from a uniform Dirichlet distribution. Then, we keep generating strings from this PCFG (starting from S and following the production rules, till it returns eos) and concatenating them till the total sequence length matches the context length $T$. To control the complexity of generated sequences and prevent infinite recursion, we impose a maximum derivation depth $d_{\max}$; once this depth is reached, expansions of the nonterminal C are restricted to produce the terminal symbol eos. For generating training sequences, we set $d_{\max} = 10$, whereas for test sequences, it is set as 0, *i.e.*, we only get strings from the set $\{\mathsf{a}\,\mathsf{b}\,\mathsf{eos}, \; \mathsf{b}\,\mathsf{a}\,\mathsf{eos}, \; \mathsf{a}\,\mathsf{a}\,\mathsf{eos}, \; \mathsf{b}\,\mathsf{b}\,\mathsf{eos}\}$ with different probabilities.

In Fig. 5, we train a transformer on sequences drawn from the two types of PCFGs. The test sequences are of the form $\{\ldots \mathsf{eos}, \mathsf{a}\}$ or $\{\ldots \mathsf{eos}, \mathsf{b}\}$. For sequences

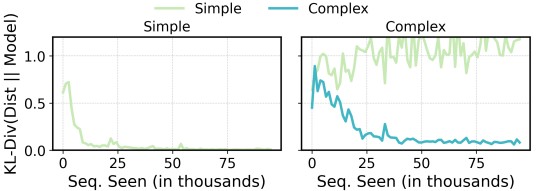

Figure 5: A transformer trained on sequences from random 'complex' and 'simple' PCFGs (see Eq. (9)) can infer the type of PCFG the context comes from. Here, the test sequences are of the form $\{\ldots, \mathsf{a}\}$ or $\{\ldots, \mathsf{b}\}$, where every string before the last element is from the set $\{\mathsf{a}\,\mathsf{b}\,\mathsf{eos}, \; \mathsf{b}\,\mathsf{a}\,\mathsf{eos}, \; \mathsf{a}\,\mathsf{a}\,\mathsf{eos}, \; \mathsf{b}\,\mathsf{b}\,\mathsf{eos}\}$. **(Left)** compares the KL divergence between model output probabilities and the probabilities of the 'simple' PCFG. **(Right)** compares the KL divergence between model output probabilities and the probabilities of the ground truth 'complex' PCFG as well as those of the 'simple' PCFG.

from the 'complex' grammar, we measure the KL divergence between the transformer's output distribution with the ground truth probabilities of the grammar, as well as those of the 'simple' PCFG. Note that since $p_3 = p_4 = 0$ for the 'simple' PCFG, $P(\mathsf{a}|\mathsf{a}) = P(\mathsf{b}|\mathsf{b}) = 0$. We

see that the transformer output distribution is closer to the 'complex' PCFG. For sequences from the 'simple' grammar, we find that the output distribution closely matches the 'simple' PCFG. This result shows that the transformer infers the type and parameters of the PCFG from which the context was generated. Here, we use the same context length for train and test sequences; please see Appendix A for test results with different context lengths.

### 3.4 Results on a Pre-trained LLM

In this section, we conduct an experiment with Boolean functions to validate in-context Occam's razor in a pre-trained LLM (GPT-4). We consider a Boolean input $\mathbf{x} \in \{0, 1\}^d$ and two Boolean functions: a 'simple' function where label $y_s = \mathbf{x}[0]$ and a 'complex' function where label $y_c = \text{Maj}(\mathbf{x}[i_1, i_2, i_3])$, where $\{i_1, i_2, i_3\} \subset [d]$ and $\text{Maj}(\mathbf{z}) = 1$ if $\sum_i \mathbf{z}[i] \geq 3/2$ and 0 otherwise, for $\mathbf{z} = \mathbf{x}[i_1, i_2, i_3]$. For instance, let $d = 5$, and $i_1, i_2, i_3$ be $1, 3, 4$, respectively. Then, some example sequence-label pairs are shown in Fig. 1.

We consider prompts with $n$ examples $(\mathbf{x}_i, y_i)$ followed by a query $\mathbf{x}_{n+1}$ in the format shown in Fig. 1. To construct these examples, we first sample the indices $i_1, i_2, i_3$ randomly from $[d]$ (this set is fixed for one prompt). We consider two types of prompts, one with ambiguous examples, where the labels predicted by both functions are the same, *i.e.*, $y_s = y_c$, to check which function the model prefers, and the other with examples from the complex function, with labels $y_c \neq y_s$, to check

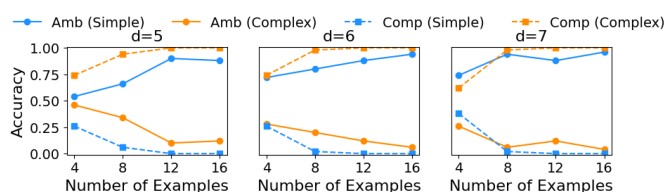

Figure 6: Results with GPT-4 for in-context learning Boolean functions with two types of prompts. We consider $y_{\text{simple}} = \mathbf{x}[0]$ and $y_{\text{complex}} = \text{Maj}(\mathbf{x}[1, 3, 4])$. Comparing accuracy with respect to the simple and complex functions on unambiguous queries, we see that the model exhibits in-context Occam's razor: it uses the simple function when given ambiguous examples and the complex function when the context can only be explained by it.

if the model can in-context learn the complex function as well. In both cases, the query is unambiguous, *i.e.*, $y_s \neq y_c$, to check whether the model uses the simple or the complex function to predict the final output.

We prompt GPT-4 with these prompts and compare the output's accuracy (where the output is the token with the largest log probability) with respect to the simple and the complex functions, over 50 prompts. The results for different numbers of examples are shown in Fig. 6. We consider $d = 5, 6, 7$. We observe that on prompts with ambiguous examples which can be explained by both the functions, the model outputs on the unambiguous query match with the simple function. On the other hand, on prompts with examples that can be explained only by the complex function, it gets higher agreement with the complex function on the query. These results confirm that GPT-4 exhibits in-context Occam's razor in this setting with Boolean functions — when two Boolean functions explain the context, the model uses the simpler one to make predictions on the query input.

## 4 Additional Experiments

In this section, we present additional results with our Markov-chain setup by (i) examining the inference behaviour of a trained transformer on sequences of varying length and (ii) testing whether LSTMs show a similar Occam's razor-like bias. Please see Appendix A for further analyses: (i) transformers trained only on the complex task do not exhibit this bias; (ii) the effect of model scale; and (iii) the effect of the training-task mixture proportion.

**Testing with Different Orders and Context Lengths.** In this section, we test how varying the context length affects a trained transformer's behaviour on sequences from different task categories. In Fig. 7, we take a transformer trained only on order-1 and order-3 chains with sequence length 400, and evaluate it on orders-1, 2, 3, and 4 sequences. For each evaluation, we measure the KL divergence of the model's output compared to uniform/ bigram/ trigram/ tetragram/ pentagram statistics for varying prompt sequence lengths.

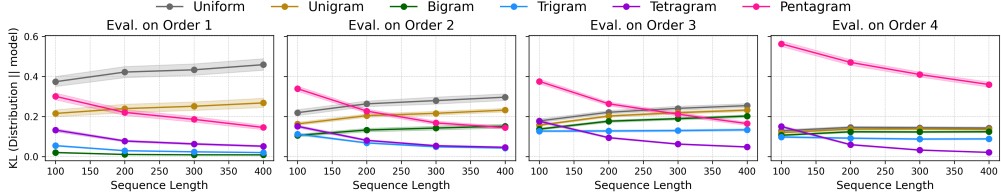

Figure 7: **Effect of context length on inference behaviour.** Figure shows a trained transformer's performance when prompted with sequences from different-order chains as a function of the sequence length. This transformer was trained on sequences from order-1 and order-3 chains with sequence length 400. Here, we also evaluate its performance on order-2 and order-4 sequences, which were not seen during training.

On order-1 sequences, KL(bigram ∥ model) stays essentially flat and low across different context lengths, indicating a consistent fit to the simplest statistic. In contrast, KL(tetragram ∥ model) is large for short contexts (where higher and lower-order statistics are less similar), and steadily drops as the context grows (as expected by Eq. (5)). The clear gap for short contexts further validates our hypothesis that the model prioritizes the simpler explanation.

Comparing results for different orders for sequence length $\geq 200$, we find that consistent with the main results in Section 3, for order-1 sequences, the KL divergence to bigram is the lowest, and for order-3, the KL to tetragram is the lowest. For order-4, we find that the model does not learn pentagram statistics, but rather returns output based on tetragram statistics. However, for order-2 sequences, which the model was never explicitly trained on, the behavior is more nuanced — while the model avoids the simplest explanation (bigram), it does not strongly commit to either the more complex explanation (tetragram) or a specific intermediate complexity it has not been directly trained on (trigram). See App. A.3 for further discussion and additional results.

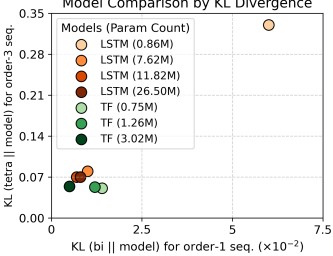

Figure 8: **Comparison with LSTMs.** Comparison between LSTMs and transformers of different capacities, trained on a mixture of order-1 and order-3 chains, and tested on unseen sequences at the end of training. We observe that LSTM models require significantly larger capacity than transformers to infer the true order (1 or 3) and predict using the corresponding statistics (bigram or tetragram) at test time.

**Results for LSTMs.** In this section, we train stacked LSTMs to investigate if they exhibit in-context Occam's razor similar to transformers. From Fig. 8, we find that LSTMs exhibit a weaker form of Occam's razor-like bias, as they require significantly more capacity to consistently select the correct underlying Markov chain order from the input sequence at inference time compared to transformers. See App. A.6 for inference results with training time for the models in Fig. 8, and App. C for training details.

## 5 Conclusion

This work demonstrates that transformers trained on hierarchical task complexity structures effectively implement a form of Bayesian Occam's razor, consistently identifying the simplest sufficient hypothesis without defaulting to more expressive model classes. Our finding raises important questions: From a mechanistic perspective, how do transformers internally implement this complexity selection? From an optimization standpoint, how do gradient descent dynamics lead to the emergence of these Bayesian selection principles during training? Future work could investigate more complex hierarchical structures beyond simple dimensionality differences, such as models with different structural constraints (e.g., various sparsity patterns in regression) or tasks where complexity varies along multiple axes simultaneously. Our findings suggest that principled hypothesis selection may be an inherent inductive bias property of transformers trained on diverse task distributions, potentially contributing to their remarkable generalization capabilities in real-world settings.

## Acknowledgments

This work was partially funded by the NSERC Discovery Grant No. 2021-03677, the Alliance Grant ALLRP 581098-22, and a CIFAR AI Catalyst Grant. PD and TB were also supported by the UBC 4YF Doctoral Fellowship. BV was supported by NSF CAREER Award CCF-2239265. This work was done in part while BV was visiting the Simons Institute for the Theory of Computing. The authors also acknowledge use of the Sockeye cluster by UBC Advanced Research Computing and the Discovery cluster by USC CARC. The authors thank Core Francisco Park for useful discussions and the anonymous reviewers for helpful feedback.

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

# Appendix

## A  Additional Experimental Results

### A.1  Effect of Curriculum Training

We used curriculum learning in the linear regression setting, following previous linear-regression ICL studies (e.g., Garg et al. (2022) and follow ups). To test its impact, we reran the experiment without any curriculum (Fig. 9, left). In that setting, the transformer eventually converges to the $d/2$-dimensional least-squares solution, but it takes substantially longer than with curriculum scheduling (Fig. 9, right).

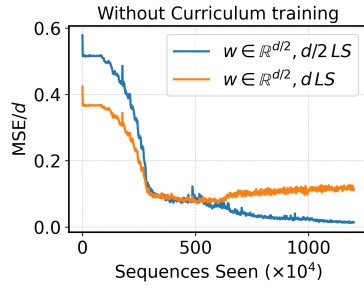
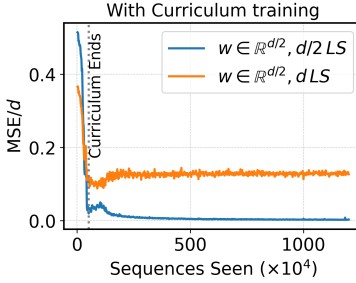

Figure 9: Normalized MSE (normalized by $d$) between the transformer's predictions and the two benchmark solutions—$d/2$-LS (simple) and $d$-LS (complex)—over training steps. Models are trained with context length $T = 40$, with $d = 20$ (same setting as Fig. 4) and evaluated for sequences with $T_{\text{test}} = 15$. **(Left)** Model trained without curriculum scheduling on equal proportions of $d/2$- and $d$-dimensional regressors in every mini-batch. **(Right)** Curriculum schedule used in Fig. 4.

Specifically, we consider the following curriculum scheduling. Training begins with sequences from $d/4$-dimensional regressors. Every 2000 steps, the dimensionality of regressors in the entire mini-batch is increased by 2 until it reaches $d/2$. From that point onward, the dimensionality of regressors in half of each batch remains at $d/2$, while the other half continues to increase by 2 every 2000 steps until it reaches $d$; thereafter, batches contain equal proportions of $d/2$ and $d$. This curriculum markedly accelerates convergence to the simpler $d/2$-LS predictor.

## A.2 Training with Only the Complex Task

In this section, we train the transformer on fixed order-$k$ Markov chains (following Edelman et al. (2024); Park et al. (2025)) to examine whether at inference time, it can infer the correct order when presented with sequences generated from a lower-order chain ($< k$). In Fig. 10, we test transformers trained on only order-3 (left) and order-2 (right) chain sequences on sequences from order-1 chains and find that the model predictions don't match bigram statistics most closely. This finding is crucial, as it suggests that a transformer trained on an order-$k$ chain learns only the order-$k$ statistics of the context and does not generalize to lower-order statistics. We repeat this experiment for the case of linear regression (see Fig. 11) and observe a similar behaviour.

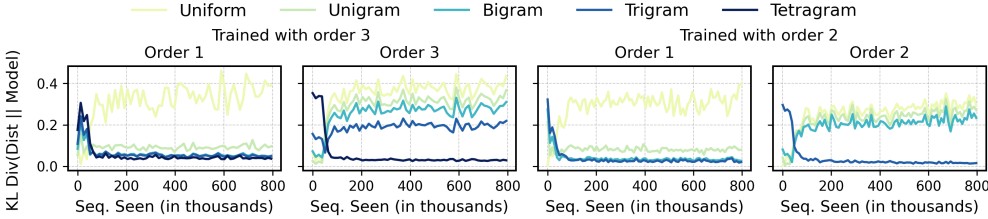

Figure 10: **Transformer trained on only the complex task does not exhibit Occam's razor-like inductive bias.** A transformer trained on random order-3 (**left column**) or order-2 (right column) Markov chains can only predict based on order-3 (left column) or order-2 (**right column**) statistics, and fails to predict based on order-1 statistics when given order-1 in-context sequences.

We also do the same experiment for the linear regression setup and train the transformer of sequences generated with only $d$-dimensional regressors, and similarly observe that the transformer does not exhibit an Occam's razor-like inductive bias.

## A.3 Testing with different context lengths

In this section, we test how varying the context length affects a trained transformer's behaviour on sequences from different task categories in the Markov chain setting and the PCFG setting.

In Fig. 7, we take a transformer trained only on order-1 and order-3 chains with sequence length 400, and evaluate it on orders-1, 2, 3, and 4 sequences. For a concrete quantification serving as a baseline for behaviors under order-2 and order-4, if we fix sequence length $= 300$, the relative gap between the lowest KL values and the second lowest (a larger gap indicates more confident prediction) are 1.618 for order-1 and 1.074 for order-3 sequences. For order-4 sequences, we find that the model does not learn pentagram statistics, but rather returns output based on tetragram statistics. Now, moving to order-2 sequences, which again the model was never explicitly trained on, the relative gap of KL distances to trigram and tetragram is 0.1073, significantly smaller than the gaps reported above for order-1 and order-3.

In Fig. 12, we evaluate the model in Fig. 3 on different context lengths at the end of training. We find that the model predictions most closely match with bigram for order-1 and trigram for order-2 sequences, respectively, as expected. We also find that the gap between bigram

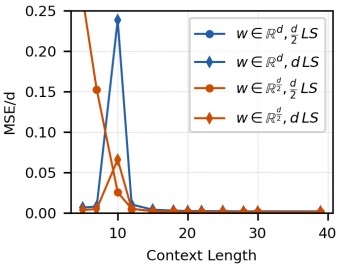

Figure 11: A transformer trained on $X = [\mathbf{x}_1, y_1, \mathbf{x}_2, y_2, ..., \mathbf{x}_T, y_T]$ sequences generated using fixed complexity, random $d$-dimensional regressors $\mathbf{w}_d$ (see Section 3 for details). We plot $(M_\theta([X, \mathbf{x}_{\text{test}}]) - \mathbf{w}_{\text{bench}}^\top \mathbf{x}_{\text{test}})^2/d$ where $\mathbf{w}_{\text{bench}}$ refers to two benchmark least-squares solutions in $d$ or $d/2$ dimensional space described in Section 3. We see that the transformer's predictions align most closely with $\mathbf{w}_d^{\text{LS}}$ no matter the type of inference-time regressor used to generate the sequences ($\mathbf{w}_d$ or $\mathbf{w}_{d/2}$). This indicates that the transformer doesn't learn to estimate the lower-complexity solution $\mathbf{w}_{d/2}^{\text{LS}}$, unlike the case in Fig. 4. Here, $T = 39$ and $d = 10$. Also the first curve (blue, dot) is out of the plotted range.

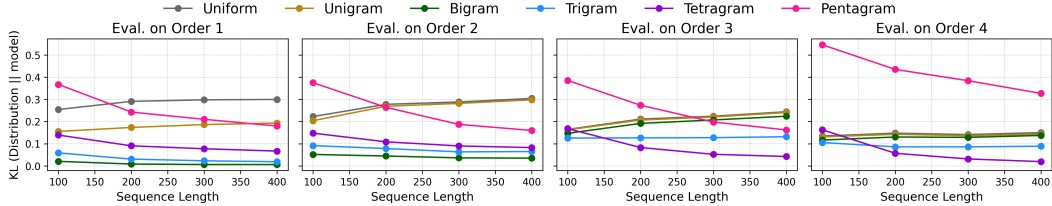

Figure 12: Figure illustrates how varying the context length affects a trained transformer's performance when prompted with sequences from different-order chains at inference. This transformer was trained on sequences from order-1 and order-2 chains with sequence length 200. The results are averaged over 15 sequences each from 250 transition matrices.

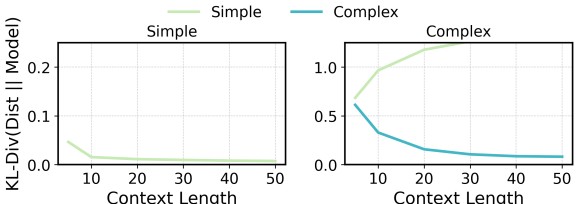

Figure 13: Figure illustrates the performance of a trained transformer when prompted with sequences of varying context lengths from the two grammars. The context length during training was 50 in this case. We find that the transformer can identify the correct type of grammar for slightly shorter context lengths, but the performance starts deteriorating for very short context lengths.

and trigram on order-1 sequences increases as context length becomes smaller. This is consistent with the observations in Fig. 7.

In Fig. 13, we evaluate the model in Fig. 5 (trained on PCFGs) on different context lengths at the end of training. We find that for both simple and complex grammars, the performance deteriorates when the context length is (very) small.

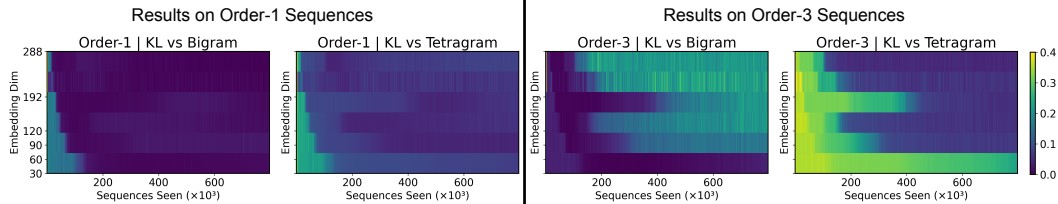

Figure 14: **Effect of model scale on inference behaviour.** Each heatmap shows the KL divergence between the model's output distribution and the indicated $n$-gram baseline as training progresses (horizontal axis), across different embedding dimensions (vertical axis). We report results for models with 6 layers, 6 heads. We observe that with embedding dimension 30, the model successfully learns bigram statistics on order-1 sequences, but fails to learn tetragram statistics on order-3 sequences. In contrast, all models with embeddimg dimension $> 30$ eventually learn both. We also observe a consistent trend: as overparameterization increases, the training time required to acquire both order-1 and order-3 statistics decreases.

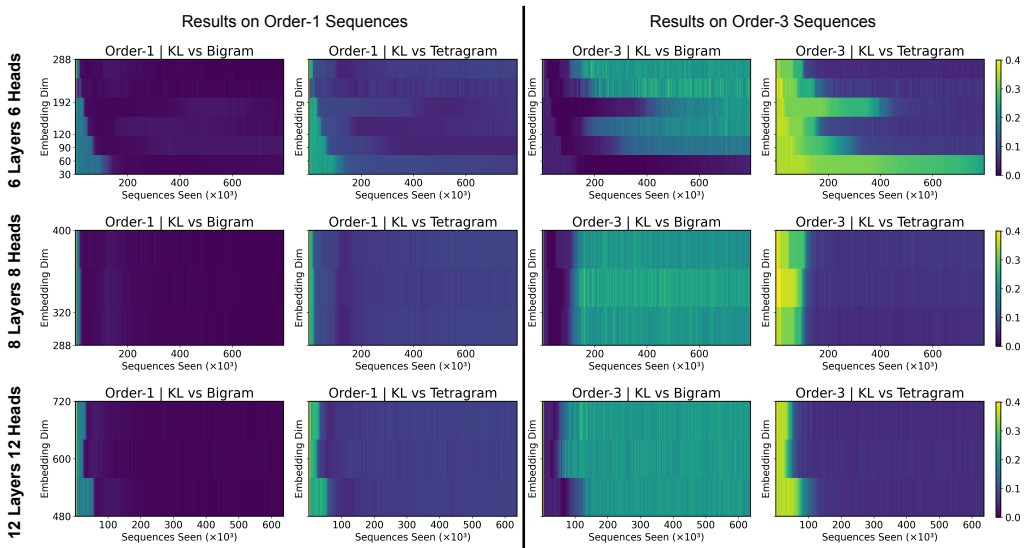

Figure 15: **Effect of model scale on inference behaviour.** Each heatmap shows the KL divergence between the model's output distribution and the indicated $n$-gram baseline as training progresses (horizontal axis), across different embedding dimensions (vertical axis). We report results for models with 6 layers, 6 heads (**top row**, used to report main results in paper), 8 layers, 8 heads **(middle row)**, and 12 layers, 12 heads **(bottom row)**. Darker/blue regions indicate lower KL divergence; brighter/yellow regions indicate higher KL. In each row, the left two heatmaps show evaluation on order-1 test sequences, and the right two on order-3 test sequences. For the 6-layer, 6-head model **(top)**, we observe that with embedding dimension 30, the model successfully learns bigram statistics on order-1 sequences, but fails to learn tetragram statistics on order-3 sequences. In contrast, all models with embeddimg dimension $> 30$ eventually learn both. We also observe a consistent trend: as overparameterization increases, the time required to acquire both order-1 and order-3 statistics decreases. This trend also appears, though less prominently, for the deeper and wider models (8L, 8H and 12L, 12H).

## A.4   Effect of Model Scale

In Fig. 14, we present results for larger transformer models by increasing the embedding dimension in a 6 layer, 6 head transformer model. We find that larger models also exhibit



Figure 16: Repetition of the experiment in Fig. 2 when training on a mixture of order-1 and order-3 chains with a 2 layer transformer, with *n* heads in the first layer, 1 head in the second layer, and embedding dimension $dim = 16n$. We observe that for order-3 sequences, the model output probabilities are not as closely aligned to tetragram statistics as seen in deeper models in Fig. 15. However, increasing the model size via scaling up the number of heads (and embedding dimension) similarly speeds up the time required to learn order-1 and order-3 statistics.

Figure 17: Repetition of the experiment in Fig. 3 when training on a mixture of order-1 and order-2 chains with a 2 layer transformer, with *n* heads in the first layer, 1 head in the second layer, and embedding dimension $dim = 16n$. We observe that increasing the model size speeds up the time required to learn order-1 and order-2 statistics, similar to the experiments with deeper models in Fig. 15 and two layer models in Fig. 16.

the Occam's razor-like inductive bias, and converge faster as the number of parameters is increased. We also observe this trend in the next few experiments where we vary the number of layers, attention heads, and the embedding dimension.

In Fig. 15 we repeat the experiments in Fig. 14 for 8 layer, 8 heads and 12 layer, 12 heads models by varying the embedding dimension. In Figs. 16 and 17, we repeat the experiments in Figs. 2 and 3, respectively, with a 2 layer transformer model, varying the embedding dimension and number of heads in the first layer, while keeping the heads in the second layer fixed to 1. We find that across all settings, the transformer infers the correct Markov chain order from the input context. We also observe that scaling the model size speeds up convergence.

## A.5 Effect of Training Mixture Proportion

In Fig. 18, we train transformers while systematically varying the fraction of order-1 and order-3 sequences in each mini-batch (order-1 fraction $\in [0.1, 0.9]$). For each setting, we evaluate at inference time: (i) KL(bigram‖model) and (ii) KL(tetragram‖model) on both order-1 and order-3 test sequences.

We find that the transformer reliably learns bigram statistics for order-1 sequences, even when order-1 examples make up a small fraction of the training mix. In contrast, learning tetragram statistics for order-3 sequences becomes increasingly difficult as the fraction of order-1 sequences rises — under such skew, the model tends to default to bigram-like behaviour. Moreover, the number of training steps required to acquire tetragram statistics increases as the training distribution becomes more imbalanced.

Increasing the batch size helps offset this effect: larger batches ensure that more order-3 sequences are seen per step in absolute terms, which raises the threshold at which the model can still learn the correct higher-order statistics.

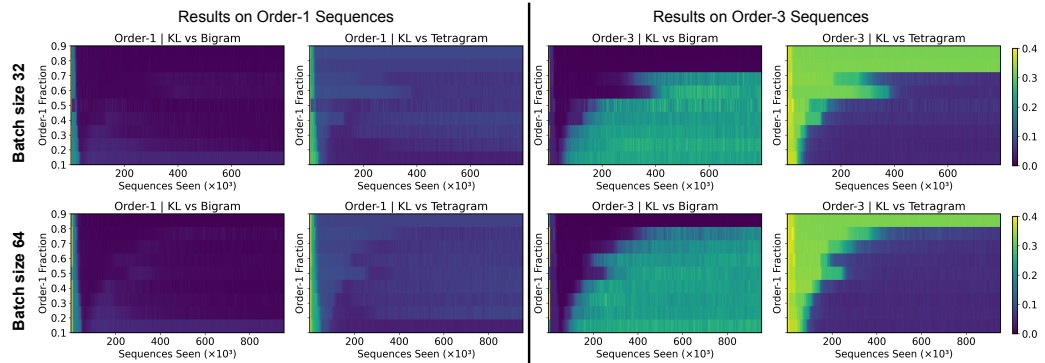

Figure 18: **Influence of training mixture proportion on inference behaviour.** Each heatmap shows the KL divergence between the model's output distribution and the indicated *n*-gram baseline as training progresses (horizontal axis), for different fractions of order-1 sequences in the mini-batches (vertical axis) using batch size 32 **(top row)** and 64 **(bottom row)**. We find that even when this fraction is very low, the model learns bigram statistics for order-1 sequences. Increasing this fraction makes the learning of tetragram statistics for order-3 sequences increasingly difficult, but increasing the batch size helps offset this effect.

### A.6 Results for LSTMs

Fig. 19 shows training time evaluation of the LSTM models shown in Fig. 8 and comparison with transformer models.

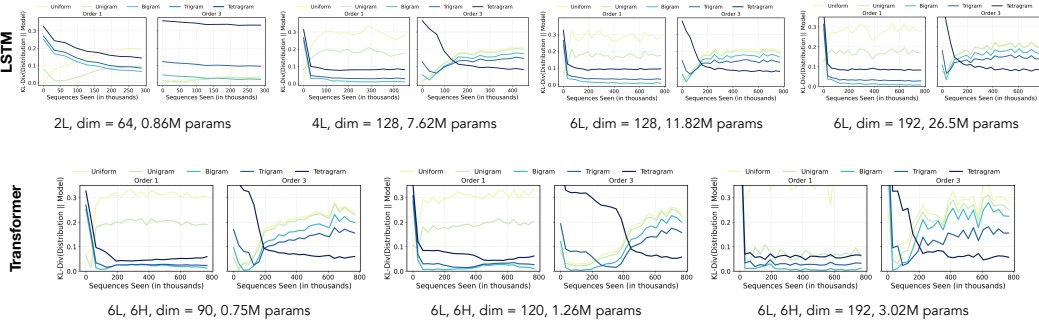

Figure 19: The plots show the KL divergence between the model's output and various *n*-gram statistics for order-1 **(left)** and order-3 **(right)** sequences, as a function of training time, across LSTM models of varying depth and width reported in Fig. 8. For direct comparison, results for a 6-layer, 6-head transformer (dim= 192) are shown in the bottom-right (reproduced from Fig. 2).

## B Additional Details for Section 3

### B.1 Markov chains: Omitted details for asymptotics of likelihood

Using a BIC style Laplace approximation to the marginal likelihood (Schwarz, 1978; Ghosal & van der Vaart, 2017; Csiszár & Talata, 2006), we have

$$\log p(X \mid s) \approx \log p(X \mid \hat{P}_s) - \frac{d_s}{2} \log T$$

Here, $\hat{P}_s$ denotes the parameters of the order-$s$ chain that maximize the likelihood of the observed sequence $X$. $d_s$ is the effective dimension of an order $s$ chain, concretely, $d_s = V^s (V - 1)$. For an order-$s$ chain, we know that $p(X \mid P_s) = \prod_t p_{P_s}(x_t \mid x_{t-1}, ..., x_{t-s})$, and

the parameters $\hat{P}_s$ that maximize this are the empirical conditional probabilities derived from the observed sequence, i.e. $p_{P_s}(x_t \mid x_{t-1}, ..., x_{t-s}) = \hat{p}(x_t \mid x_{t-1}, ..., x_{t-s})$ for each context of length $s$. Formally, $\hat{p}_X(x_t \mid x_{t-1}, ..., x_{t-s}) = \frac{n_{x_{t-s}, ..., x_{t-1}, x_t}(X)}{n_{x_{t-s}, ..., x_{t-1}}(X)}$, where $n_{x_i, ..., x_j}(X)$ denotes the number of symbol transitions $x_i \to ... \to x_j$ in the sequence $X$.

## B.2 Linear Regression: Omitted training details

Training sequences with $T$ in-context examples are generated as follows: we first sample $s$ uniformly from the set $\dim = \{d/2, d\}$, then generate $\mathbf{w}_s \sim \mathcal{N}(\mathbf{0}, \mathbb{I}_s)$. For task dimension $s = d/2$, we effectively use $\mathbf{w} = [\mathbf{w}_s, \mathbf{0}]$ (zero-padded to dimension $d$). We then sample feature vectors $\mathbf{x}_t \sim \mathcal{N}(\mathbf{0}, \mathbb{I}_d)$ and generate the labels $y_t = \mathbf{w}^\top \mathbf{x}_t$ for $t \in [T]$, forming sequences $X_{\leq t} = [\mathbf{x}_1, y_1, \mathbf{x}_2, y_2, ..., \mathbf{x}_t, y_t]$. We use this to train a transformer $M_\theta$ to auto-regressively predict the label $y_{t+1}$ using the first $t$ pairs of examples $X_{\leq t}$ by minimizing the population square-loss (analogous to Eqn. (3)).

$$L(\theta) := \mathbb{E}_{\substack{\mathbf{x}_{1:T} \sim \mathcal{N}(\mathbf{0}, \mathbb{I}_d) \\ \mathbf{w}_s \sim \mathcal{N}(\mathbf{0}, \mathbb{I}_s) \\ s \sim \mathrm{Unif}(\dim)}} \sum_{t=1}^{T} \ell(M_\theta(X_{\leq t}; \mathbf{x}_{\text{test}}), y_{t+1}), \tag{10}$$

where $\ell$ is the squared loss.

## B.3 Bayes' Optimal for Linear Regression

We can explain the preference for the simpler hypothesis using a similar Bayesian viewpoint as done in Section 3.1 for Markov chains. Specifically, the Bayes' optimal prediction on the test query $\mathbf{x}_{\text{test}}$ given the in-context demonstrations is $y = \mathbf{w}_{\text{BO}}^\top \mathbf{x}_{\text{test}}$, where $\mathbf{w}_{\text{BO}}$ is the mixture: $\mathbf{w}_{\text{BO}} = \sum_{d' \in \{d/2, d\}} p(d' \mid A_d, \mathbf{y}) \cdot \tilde{\mathbf{w}}_{d'}^{\text{LS}}$. Here, $p(d' \mid A_d, \mathbf{y})$ denotes the posterior probability of the $d'$-dimensional regressor given the matrix vector $A_d$ and the labels $\mathbf{y}$, and $\tilde{\mathbf{w}}_{d'}^{\text{LS}} = [\mathbf{w}_{d'}^{\text{LS}}, \mathbf{0}]$. Using Bayes' theorem, we express the posterior probabilities in terms of the likelihoods (assuming a uniform prior on regressor dimensionality): $p(d' \mid A_d, \mathbf{y}) = p(\mathbf{y} \mid A_d, d') / \sum_{r \in \{d, d/2\}} p(\mathbf{y} \mid A_d, r)$. For Gaussian priors over $\mathbf{w}_{d'}$, we can derive analytical expressions for these likelihoods. Similar to our Markov chain analysis, the likelihood incorporates a complexity penalty that favors lower-dimensional models. Consequently, when the underlying regressor generating sequence $X$ is $\mathbf{w}^* = \mathbf{w}_{d/2}$ and $d > T > d/2$, we find that $p(d/2 \mid A_d, \mathbf{y}) \gg p(d \mid A_d, \mathbf{y})$, resulting in $\mathbf{w}_{\text{BO}} \approx \mathbf{w}_{d/2}^{\text{LS}}$. Conversely, when $\mathbf{w}^* = \mathbf{w}_d$, the improved fit outweighs the complexity penalty, giving us $\mathbf{w}_{\text{BO}} = \mathbf{w}_d^{\text{LS}}$. This Bayesian mechanism explains why transformers implement Occam's razor by selecting the simplest hypothesis in-context.

**Calculating the Bayes Optimal.** In order to find the Bayes optimal, which is the posterior mean of $\mathbf{w}$ conditioned on the sequence $X$, we need the conditional distribution $p(\mathbf{w} \mid A_d, \mathbf{y})$. By Bayes' rule,

$$p(\mathbf{w} \mid A_d, \mathbf{y}) = \frac{p(\mathbf{y} \mid A_d, \mathbf{w}) \, p(\mathbf{w})}{\int p(\mathbf{y} \mid X, \mathbf{w}') \, p(\mathbf{w}') \, d\mathbf{w}'}. \tag{11}$$

Under uniform prior on the regressor dimensionality, the overall prior $p(\mathbf{w}) = \frac{1}{2} p_{d/2}(\mathbf{w}) + \frac{1}{2} p_d(\mathbf{w})$, where $p_{d/2}(\mathbf{w}), p_d(\mathbf{w}) \sim \mathcal{N}(\mathbf{0}, \mathbb{I}_d)$ denote the respective priors over the $d/2$ and $d$ dimensional regressors. Here, $p_{d/2}(\mathbf{w})$ corresponds to a Gaussian $\mathcal{N}(\mathbf{0}, \mathbb{I}_{d/2})$ over the first $d/2$ components of $\mathbf{w}$ with zeroes in the remaining $d/2$. Plugging this into Eq. (11), and using the fact the $\mathbf{w}_{d/2}^{\text{LS}}$ and $\mathbf{w}_d^{\text{LS}}$ are the Bayes' optimal solutions under their respective priors, we get a mixture of solutions

$$\mathbb{E}[\mathbf{w} \mid A_d, \mathbf{y}] = \frac{L_{d/2}}{L_{d/2} + L_d} \tilde{\mathbf{w}}_{d/2}^{\text{LS}} + \frac{L_d}{L_{d/2} + L_d} \tilde{\mathbf{w}}_d^{\text{LS}},$$

where $L_{d'} = p(\mathbf{y}|A_d, d')$ denote the marginal likelihood of $\mathbf{y}$ under the regressor of dimensionality $d'$. Notice the term $\frac{L_{d'}}{\sum_r L_r} = p(d'|A_d, \mathbf{y})$ using Bayes' rule and uniform prior over the dimensionality of regressor. Thus, the Bayes' optimal is the following posterior-weighted combination of the respective LS solutions

$$\mathbf{w}_{\text{BO}} = \sum_{d' \in \{d/2,\, d\}} p(d' \mid A_d, \mathbf{y}) \cdot \tilde{\mathbf{w}}_{d'}^{\text{LS}}.$$

**Expressions for $L_1$ and $L_2$.** The marginal likelihood $L_{d'}$ under the noiseless linear regression setting is

$$L_{d'} = \int \delta(\mathbf{y} - A_d \mathbf{w}) \, p_{d'}(\mathbf{w}) \, d\mathbf{w}.$$

This can be interpreted as the "density" of $\mathbf{y}$ when viewed as a random variable $A_d \mathbf{w}$ for $\mathbf{w} \sim p_{d'}$. Concretely, if the sequence generating regressor $\mathbf{w}^* \sim p_d(\mathbf{w})$, then $L_{d/2} = 0$ (a.s.), as no $\mathbf{w} \sim p_{d/2}$ solves $\mathbf{y} = A_d\mathbf{w}$. Hence, $\mathbf{w}_{\text{BO}} = \mathbf{w}_d^{\text{LS}}$.

**Bayesian Occam's Razor.** Now consider the other case when $\mathbf{w}^* \sim p_{d/2}(\mathbf{w})$. Here, regressors from either prior can "explain" the context and will have non-zero marginal likelihood. Consider the regime $d > T \geq d/2$, the density of $\mathbf{y}$ under $p_{d/2}$ is

$$L_{d/2} = \frac{1}{(2\pi)^{d/4}\sqrt{\det(A_{d/2}^\top A_{d/2})}} \exp\left(-\tfrac{1}{2}\left\|\mathbf{w}_{d/2}^{\text{LS}}\right\|^2\right),$$

$$L_d = \frac{1}{(2\pi)^{T/2}\sqrt{\det(A_d A_d^\top)}} \exp\left(-\tfrac{1}{2}\left\|\mathbf{w}_d^{\text{LS}}\right\|^2\right).$$

In order to find the ratio of the above marginal likelihoods, we need to compute the difference

$$\Delta_d := \log\det(A_d A_d^\top) - \log\det(A_{d/2}^\top A_{d/2}).$$

In order to compute $\Delta_d$, we first look at its expectation. Later, we use a concentration argument to get a high probability bound on $\Delta_d$.

**Expectation of $\Delta_d$.** As $A_d$ has i.i.d. Gaussian entries, both $A_d A_d^\top$ and $A_{d/2}^\top A_{d/2}$ follow Wishart distributions. It is known (e.g., Zwiernik et al. (2016, in Prop. A.1)) that

$$\mathbb{E}\log\det(A_d A_d^\top) = \psi_T(d/2) + T\log 2,$$

$$\mathbb{E}\log\det(A_{d/2}^\top A_{d/2}) = \psi_{d/2}(T/2) + (d/2)\log 2,$$

where $\psi_p$ is the multivariate digamma function. InWe will first calculate the difference It further holds that

$$\psi_T(d/2) - \psi_{d/2}(T/2) = \sum_{i=1}^{T} \psi\left(\frac{d+1-i}{2}\right) - \sum_{j=1}^{d/2} \psi\left(\frac{T+1-j}{2}\right), \tag{12}$$

where $\psi$ is the digamma function. The asymptotic expansion of the digamma function gives us

$$\psi(z) = \log(z) - \tfrac{1}{2z} + O(z^{-2}), \qquad |arg(z)| < \pi, z \to \infty. \tag{13}$$

Let $z_i = d + 1 - i$ and $z_j' = T + 1 - j$. Using Eq. (13) in Eq. (12) when $T, d \to \infty$, and ignoring the second order terms, we have

$$\psi_T(d/2) - \psi_{d/2}(T/2) = \sum_{i=1}^{T}\left[\log(z_i) - \tfrac{1}{2z_i}\right] - \sum_{j=1}^{d/2}\left[\log(z_j' - \tfrac{1}{2z_j'})\right] - (T - d/2)\log 2$$

$$= \underbrace{\log\left(\frac{\prod_{i=1}^{T} z_i}{\prod_{j=1}^{d/2} z_j'}\right)}_{\text{Term-1}} + \underbrace{\frac{1}{2}\sum_{j=1}^{d/2}\frac{1}{z_j'} - \frac{1}{2}\sum_{i=1}^{T}\frac{1}{z_i}}_{\text{Term-2}} - (T - d/2)\log 2 \tag{14}$$

Term-1 can be calculated as

$$\log\left(\frac{\prod_{i=1}^{T} z_i}{\prod_{j=1}^{d/2} z'_j}\right) = \log\left(\frac{d!(T-d/2)!}{T!(d-T)!}\right)$$

$$= d\log(d) - d + (T-d/2)\log(T-d/2) - (T-d/2) - T\log(T) + T - (d-T)\log(d-T) + d - T$$

$$= (d\log d)(c-1/2) + d\underbrace{\left((c-1/2)(\log(c-1/2)-1) - c\log c - (1-c)\log(1-c)\right)}_{g(c)},$$

where the second equality follows by using Stirling's approximation, and neglecting the logarithmic terms. The third equality follows by using $T = cd$, where $c \in (1/2, 1)$. It can be shown that $g(c)$ is a bounded function in $c \in (1/2, 1)$. We can easily see that it is a monotonically decreasing function

$$g'(c) = \log\left(\frac{(c-1/2)(1-c)}{c}\right) < 0, \quad c \in (1/2, 1).$$

Using this we have, $g(1) \le g(c) \le g(1/2)$, where $g(1/2) = \log 2$ and $g(1) = -\frac{1}{2}(\log 2 + 1)$. Therefore, we can write Term-1 as

$$\log\left(\frac{\prod_{i=1}^{T} z_i}{\prod_{j=1}^{d/2} z'_j}\right) = (d\log d)(c-1/2) + \Theta(d). \tag{15}$$

Term-2 can be bounded as

$$\frac{1}{2}\sum_{j=1}^{d/2}\frac{1}{z'_j} - \frac{1}{2}\sum_{i=1}^{T}\frac{1}{z_i} \le \frac{1}{2}\frac{d}{2}\frac{1}{T+1-d/2} - \frac{1}{2}\frac{T}{d} = \frac{1}{2}\left(\frac{1/2}{c-1/2+1/d} - c\right) = O(1),$$

$$\frac{1}{2}\sum_{j=1}^{d/2}\frac{1}{z'_j} - \frac{1}{2}\sum_{i=1}^{T}\frac{1}{z_i} \ge \frac{1}{2}\frac{d}{2}\frac{1}{T} - \frac{1}{2}\frac{T}{d+1-T} = \frac{1}{2}\left(\frac{1}{2c} - \frac{c}{1-c+1/d}\right) = o(1),$$

where we use $T = cd$, where $c \in (1/2, 1)$.

We can simplify Eq. (14) by using Eq. (15) with only the dominant $d\log d$ term and ignoring $\Theta(1)$ Term-2. Then, using this difference in the multivariate digamma functions, we have

$$\mathbb{E}[\Delta_d] = \mathbb{E}\log\det(A_d A_d^\top) - \mathbb{E}\log\det(A_{d/2}^\top A_{d/2}) = \Theta(d\log d). \tag{16}$$

**High probability bound on $\Delta_d$.** Cai et al. (2015, Thm. 1) establish a *central-limit theorem* (CLT) for the log–determinant of a high-dimensional sample covariance matrix. Specialised to our (un-normalised) Gram matrix $A_d A_d^\top$, their result gives

$$\frac{\log\det(A_d A_d^\top) - \mathbb{E}[\log\det(A_d A_d^\top)]}{\sigma_{d,T}} \xrightarrow{d\to\infty} \mathcal{N}(0,1), \qquad \sigma_{d,T}^2 = \sum_{i=1}^{d}\psi_1\left(\frac{T-i+1}{2}\right),$$

where $\psi_1$ is the trigamma function. Exactly the same argument applied to the $A_{d/2}^\top A_{d/2}$ yields

$$\frac{\log\det(A_{d/2}^\top A_{d/2}) - \mathbb{E}[\log\det(A_{d/2}^\top A_{d/2})]}{\sigma_{T,d/2}} \xrightarrow{T\to\infty} \mathcal{N}(0,1), \qquad \sigma_{T,d/2}^2 = \sum_{j=1}^{d/2}\psi_1\left(\frac{T-j+1}{2}\right).$$

Each centred log–determinant is sub-Gaussian with variance proxy $\sigma_{d,T}, \sigma_{T,d/2} = O(1)$. More concretely,

$$\Pr\left(\left|\log\det(A_d A_d^\top) - \mathbb{E}\log\det(A_d A_d^\top)\right| > t\right) \le 2\exp(-c\,t^2), \quad \text{and likewise for } A_{d/2}A.$$

Choosing $t = C\log d$ and applying a union bound gives, for a suitable constant $C > 0$,

$$\Pr\left(\left\{\left|\log\det(A_d A_d^\top) - \mathbb{E}\log\det(A_d A_d^\top)\right| \le C\log d\right\}\right.$$

$$\left. \text{and } \left\{\left|\log\det(A_{d/2}^\top A_{d/2}) - \mathbb{E}\log\det(A_{d/2}^\top A_{d/2})\right| \le C\log d\right\}\right) \ge 1 - \frac{2}{d}.$$

Hence, the difference satisfies

$$\left| \Delta_d - \mathbb{E}[\Delta_d] \right| \ \leq \ 2C \log d \quad \text{with probability } 1 - \tfrac{2}{d}. \tag{17}$$

Combining Eq. (17) with the expectation difference computed in Eq. (16) yields

$$\frac{\det(A_d A_d^\top)}{\det(A_{d/2}^\top A_{d/2})} \ = \ \exp\big(\mathbb{E}[\Delta_d]\big) \ \exp\big(O(\log d)\big) \ \geq \ d^{c'd},$$

with probability at least $1 - 2/d$ and some constant $c' > 0$ which is dependent on the ratio $c = T/d \in (\frac{1}{2}, 1)$.

At the same time, it is easy to show that $\|\mathbf{w}_d^{\text{LS}}\|^2 - \|\mathbf{w}_{d/2}^{\text{LS}}\|^2 > 0$. Put together, we have

$$L_{d/2}/L_d \gtrsim (2\pi)^{T/2 - d/4} d^{c'd}$$

which is $\gg 1$ for large $d$ and $T > d/2$. This implies that, $p(d/2 \mid A_d, \mathbf{y}) \approx 1$, and $\mathbf{w}_{\text{BO}} \approx \mathbf{w}_{d/2}^{\text{LS}}$.

### B.4 PCFGs

Formally, a PCFG is described by the tuple $\{\mathsf{N}, \mathsf{T}, \mathsf{R}, \mathsf{S}, \pi\}$, where $\mathsf{N}$ is a set of nonterminal symbols, $\mathsf{T}$ is a set of terminal symbols, and $\mathsf{R}$ is a collection of production rules that dictate how nonterminals are expanded into sequences of terminals and nonterminals. A special start symbol $\mathsf{S}$ initiates the generation process, and $\pi$ assigns probabilities to these production rules, indicating their likelihood of selection during sequence generation.

## C Details of Experimental Settings and Code

For experiments on Markov chains, linear regression and PCFGs, we train GPT-2 type decoder-only transformer (Karpathy, 2023). We use AdamW (Loshchilov & Hutter, 2019) optimizer in all experiments with a learning rate of $10^{-4}$, unless stated otherwise. We set the batch size as 32.

**Markov chains.** For all Markov chain experiments, vocab size $V = 3$. For the order-1 and order-3 experiments in Fig. 2, context length $T$ for all sequences was set 300. We used a 6 layer, 6 head transformer, with embedding dimension set to 192. For the order-1 and order-2 experiments in Fig. 3, context length $T$ was set to 200. We used a 2 layer transformer with 20 heads, and embedding dimension was set to 320. Additionally, we used relative position encoding in all the experiments. For the fixed-order experiments in Fig. 10, we used the setup from the corresponding variable order experiment.

**Linear regression.** We used a 12 layer, 8 head transformer with embedding dimension 256 similar to Garg et al. (2022). We set $d = 10$ and 20 (for Fig. 4), and context length $T = 39$.

**PCFGs.** We used a 4 layer, 4 head transformer with embedding dimension 128, and context length of 50.

**LSTMs.** All LSTM models were trained with the AdamW optimizer (learning rate $2 \times 10^{-3}$, $\beta$s equal to (0.9, 0.99)) and cosine learning rate decay with a minimum learning rate of $10^{-5}$. In all experiments, we set hidden dimension to be 4 times the embedding dimension.

**Evaluation details.** The results in Fig. 7 and Fig. 12 are averaged over 15 sequences each from 250 transition matrices reported with 95% bootstrapped confidence intervals. The results in Fig. 4, Fig. 11 are averaged across 2000 contexts, each generated with an independently sampled ground-truth regressor. Shaded bands show 95% confidence intervals obtained from 100 bootstrap trials.

**Code.** The code is available at https://github.com/puneesh00/ICL-Bayes-Occam.git

# D  Construction for Markov Chains of Varying Complexity

Recall from Eq. (6) that the model learns tasks of different complexity by implementing the Bayes optimal strategy. To achieve this, the model must compute several key quantities, including the $s$-gram statistics $p(\cdot \mid x_T, \ldots, x_{T-s+1})$ and the category posterior.

Previous work has shown that a two-layer attention-only model with $s$ heads in the first layer can compute the relevant $s$-gram statistics for the last token (Edelman et al., 2024; Rajaraman et al., 2024). To extend those results, for order one statistics, we demonstrate that by using $V$ heads in the second layer, the model can represent all the conditional distributions $p(u \mid v)$ for any pair of vocabulary tokens simultaneously. These conditional distributions are the building blocks that produce the log-likelihood used to compute the posterior. This approach can be extended to higher-order Markov chains by using $V^s$ heads in the second layer.

**Lemma 1.** *A two-layer attention-only transformer with one attention head in the first layer, and $V$ attention heads in the second layer can output* all *the empirical conditional distributions*

$$p(u \mid v) = \frac{\sum_{i=2}^{T} \mathbb{1}(x_{i-1} = v, x_i = u)}{\sum_{i=2}^{T} \mathbb{1}(x_{i-1} = v)} \quad \text{for any } u, v \in [V].$$

*for any input sequence X.*

*Proof.* We outline how to set the transformer's parameters so that, for each pair $(v, u)$, the model's final output can encode the conditional probability $p(u \mid v)$.

Recall that the input to the model is a sequence $X = [x_1, \ldots, x_T]$ of length $T$ with $x_t \in [V]$. Let $z_t \in \mathbb{R}^d$ be the input embedding of the $t$-th element of the sequence $x_t$, and $Z = [z_1, z_2, \ldots, z_T]_{T \times d}^{\top}$ be the sequence of input embeddings.

We write the two-layer transformer with $V$ heads in the second layer as

$$f(X) = W_o \left[ h_1(Z) \parallel \cdots \parallel h_V(Z) \right],$$

where "$\parallel$" denotes concatenation, $W_o \in \mathbb{R}^{V^2 \times dV}$ is the final linear projection, and each $h_v(Z)$ is the output of the $v$-th attention head in the second layer. Specifically,

$$h_v(Z) := (\text{Attn}_2^v + I) \circ (\text{Attn}_1^v + I)(Z),$$

with

$$\text{Attn}(Z) = \text{softmax}(\text{mask}(A)) Z W_V,$$

where $W_V$ is the value matrix, and the attention map $A \in \mathbb{R}^{T \times T}$ is defined as

$$A_{i,j} = \frac{(z_i^{\top} W_Q + r_{i-j+1}^{\top}) W_K z_j^{\top}}{\sqrt{d}}.$$

Here $W_Q$, $W_K$, are the key and query matrix, $r_k$ is the relative positional embedding, and $\text{mask}(\cdot)$ enforces causal masking so that position $i$ can only attend to positions $1, \ldots, i$.

We will show how to choose the model's weights so that, for each $v$, the embedding from the $v$-th head of the second layer, $h_v(Z)$, produces vectors which—when multiplied by $W_o$—yield the conditional distribution $p(\cdot \mid v)$ across the sequence $X$. For simplicity, we will omit the superscript $v$ in $\text{Attn}_\ell^v$ from now on.

**Layer 1: Isolating the previous token.** We first show how to configure the weights of the first layer so that each position $i$ copies the immediately preceding token $z_{i-1}$. For this construction we need the embedding dimension of $d = 3V$.

Set the embedding matrix $E = [I_V, I_V, 0]_{V \times d}$. In this case, the input embedding $z_t$ is $= [\underbrace{z_t^1}_{V}^\top, \underbrace{z_t^2}_{V}^\top, \underbrace{z_t^3}_{V}^\top]^\top = [e_{x_t}^\top, e_{x_t}^\top, 0]^\top$ where $e_v$ is the $v$-th basis vector.

Now let the key, query matrix and the positional embeddings be as follows.

$$W_Q = 0, \quad W_K = \begin{pmatrix} c\, I_{V \times V} & 0 & 0 \\ 0 & 0 & 0 \\ 0 & 0 & 0 \end{pmatrix}, \quad R = e_2 \cdot 1^\top,$$

where $c$ is a large constant. With these weights, the attention matrix $A$ is zero for all entries except when $j = i - 1$; that is, $A_{i,j} = 0$ if $j \neq i - 1$. For the entry where $j = i - 1$, we have $A_{i,i-1} = 1^\top W_K z_{i-1} = c$. As $c$ approaches infinity, the softmax function saturates. In other words, if we define

$$S = \text{softmax}\big(\text{mask}(A)\big),$$

then $S_{i,j} = 1$ when $j = i - 1$ and 0 for all other $j$. This ensures that the model only attends to the previous position in the first layer.

Now choosing

$$W_V = \begin{pmatrix} 0 & 0 & 0 \\ 0 & I_{V \times V} & 0 \\ 0 & 0 & 0 \end{pmatrix},$$

we have

$$\text{Attn}_1(Z)_{i,:} = \sum_{t=1}^T S_{i,t} z_t^\top W_V = [0, {z_{i-1}^2}^\top, 0]^\top.$$

This extracts and carries along the "middle" $V$ components of the embedding at position $i - 1$. Putting things together, the embedding of after the first layer will be

$$z_i' := (\text{Attn}_1 + I)(Z)_i = \begin{bmatrix} z_i^1, & z_i^2 + z_{i-1}^2, & z_i^3 \end{bmatrix}.$$

**Layer 2 - $v$-th head: Computing conditional probabilities** $p(\cdot \mid v)$**.** We now use $V$ heads in the second layer to encode, for each possible previous token $v \in V$, how likely the next token is $u \in V$.

In this layer, we set the relative positional embedding $r = 0$ and

$$W_Q = \begin{pmatrix} 0 & 0 & 0 \\ 0 & 1_v e_v^\top & 0 \\ 0 & 0 & 0 \end{pmatrix}, \quad W_K = c \begin{pmatrix} 0 & 0 & 0 \\ -e_v e_v^\top & e_v e_v^\top & 0 \\ 0 & 0 & 0 \end{pmatrix}.$$

With this weight configuration, we have

$$A_{T,i} = {z_T'}^\top W_Q W_k^\top z_i' = \begin{cases} c & \text{if } x_{i-1} = v \\ 0 & \text{o.w.} \end{cases}$$

Then, as $c \to \infty$,

$$S_{T,i} = \frac{\mathbb{1}(x_{i-1} = v)}{\sum_{t=1}^T \mathbb{1}(x_{t-1} = v)}$$

Now setting

$$W_V = \begin{pmatrix} 0 & 0 & I_V \\ 0 & 0 & 0 \\ 0 & 0 & 0 \end{pmatrix},$$

we have

$$h_v(Z) := \sum_{t=1}^{T} S_{T,t} W_V^\top z_t' + z_T' = \sum_{u \in [V]} \frac{\sum \mathbb{1}(x_{i-1} = v, x_i = u)}{\sum \mathbb{1}(x_{i-1} = v)} [0, 0, e_u^\top]^\top + z_T'$$

$$= \sum_{u \in [V]} p(u \mid v)[0, 0, e_u^\top]^\top + z_T'$$

Finally, with $P_{V \times 3V} = [0, 0, I]$, setting $W_O$ as

$$W_O = \begin{pmatrix} P & 0 & \cdots & 0 \\ 0 & P & \cdots & 0 \\ 0 & 0 & \cdots & 0 \\ 0 & 0 & \cdots & P \end{pmatrix},$$

we have

$$f(Z) = W_O[\, h_1(Z) \parallel \cdots \parallel h_V(Z)\,]$$
$$= [\, \sum_u p(\cdot \mid v = 1)e_u \parallel \cdots \parallel \sum_u p(\cdot \mid v = V)e_u\,].$$

$\square$

# E  Related Work

**Task Diversity: Learning vs Retrieval.**   In the setups investigated by Garg et al. (2022); Edelman et al. (2024), the transformer is trained on sequences generated from fresh tasks drawn from the appropriate distributions at each optimization iteration. Raventos et al. (2023); Park et al. (2025) have further extended this study for linear regression and Markov chains respectively, examining scenarios where the model is trained on sequences generated from a finite number of tasks, thus modeling and analyzing the impact of task diversity in ICL and revealing a tradeoff between two distinct modes of ICL: task retrieval and task learning (Pan et al., 2023). See also Lu et al. (2025) for a high-dimensional statistical approach that theoretically justifies some of these empirical findings. In this work, we focus on the learning mechanism of ICL and maintain the vanilla setting of training with fresh tasks at each iteration. However, in our settings these tasks originate from a finite number of complexity categories rather than a single one.

**Theoretical Explanations: Linear Regression and Markov Chains.**   One of the first works investigating the theoretical foundations of ICL is Xie et al. (2022), which proposed a Bayesian perspective that has informed numerous subsequent studies, e.g., Raventos et al. (2023). Most relevant to our work, Edelman et al. (2024) demonstrated that the bigram statistics rule implemented by transformers trained on order-1 Markov chains is the Bayes' optimal solution (given the context) under the Dirichlet prior. Similarly, for linear regression, the least-squares solution learned by transformers is the Bayes' optimal predictor (given the context) under Gaussian priors in the setting of Garg et al. (2022). A complementary approach to explaining ICL has focused on explicit weight constructions that implement functionalities observed to facilitate ICL, such as gradient descent on the context for linear regression (Li et al., 2024; 2023; Ahn et al., 2023; von Oswald et al., 2022; Fu et al., 2024) and statistical induction heads for Markov chains (Edelman et al., 2024; Rajaraman et al., 2024; Chen et al., 2024b). However, relatively few studies have investigated how—or whether—these weight configurations can actually be reached during training (Zhang et al., 2023; 2024; Chen et al., 2024a; Zhang et al., 2025). None of these theoretical frameworks has addressed the specific setting of multiple task categories with hierarchical complexity relationships. In this work, we extend the Bayesian viewpoint to explain how transformers select the simplest sufficient hypothesis in both linear regression and Markov chain settings. While these two domains differ in several respects—the format of the prompt (pairs of inputs vs. stream of symbols), the optimization objective (squared loss vs. next-token prediction), and the underlying mechanisms (gradient descent vs. statistical induction heads)—we investigate both settings under a unifying prism.

**Algorithm Selection.** Most closely related to our work are Lin & Lee (2024); Bai et al. (2023) investigating ICL regression across multiple hypotheses. Lin & Lee (2024) considers ICL of linear regression where features $\mathbf{x}$ and tasks $\mathbf{w}$ are drawn from Gaussian mixtures, generalizing the setting of Garg et al. (2022). This richer framework enables a principled study of the task retrieval versus learning modes through a Bayesian perspective. Similarly, Bai et al. (2023) investigates ICL under tasks from multiple categories, such as linear regression, noisy linear regression, and linear classification. They demonstrate that transformers implement algorithm selection in context, applying the most appropriate algorithm for the provided context at inference time. However, these works do not address tasks from different categories with clearly defined hierarchical complexity relationships, where a higher-complexity category can fully explain the context generated by simpler categories. In this particularly challenging setting, we show that algorithm selection is implemented via a form of Bayesian Occam's razor, allowing transformers to identify and apply the simplest sufficient hypothesis. We demonstrate this both for regression and Markov-chain settings. More recent work by Elmoznino et al. (2025) posits that a meta objective of ICL is linked to a prequential coding algorithm that simultaneously minimizes both algorithm complexity and prediction error on in-context demonstrations. While this intriguing interpretation reveals a form of simplicity preference in ICL, it differs fundamentally from our focus on hypothesis selection across explicitly defined hierarchical complexity structures. Additionally, Xiong et al. (2024) explore how pretrained LLMs perform ICL when several tasks are presented in superposition within a single context. Not only does their framework differ by embedding multiple tasks in the same context simultaneously, but also these tasks are not organized into hierarchically related complexity classes.

