# OpenReview forum: "In-Context Occam’s Razor: How Transformers Prefer Simpler Hypotheses on the Fly"
_colmweb.org/COLM/2025/Conference — COLM 2025_

### Official Review · Reviewer_neZS · 2025-05-13

**Rating:** 7
**Confidence:** 3
**Ethics Flag:** 1

**Summary:**

The paper studies the performance of in-context-learning in transformers on tasks with varying complexity levels. It investigates if a simpler solution is preferred for a simple task even if the transformer  has access to a more complex solution. There are both theoretical and empirical arguments presented to conclude that indeed this is the case.

The markov-chain experiment’s results do not seem to support the hypothesis clearly, but are not presented as such and thus I recommend rejection.

**Questions To Authors:**

- In figure 2, the legend and plots do not match up. Does the lime yellow plot correspond to uniform or tetragram?
- What are the results of the markov chain experiment on order-$k’$ sequences where $1<k’<k$?

**Reasons To Accept:**

- The paper presents a novel hypothesis and proposes clearexperiments to test it.
- It conducts precise experiments in three different settings (markov chain, linear regression, and synthetic grammars) to study the claims.
- There is well-written theory to support the empirical evidence.

**Reasons To Reject:**

The markov chain experiment is the expanded on and studied the most in the main body of the paper, but the results seem to not clearly support the claim that on order-1 chains the model resorts to using the simpler hypothesis.

In both Figure 1 and 2, it is correct that the model’s output has a clear separation between the order-1 and order-$k$ statistics on the higher order sequence and chooses the more complex solution. But on the left side for the simpler lower order sequences, the KL divergence is equal for both the order-1 and order-$k$ statistics. This makes sense because in equation 5, the order-$k$ statistics calculated on an order-1 sequence (for a long enough sequence or enough samples), should be equal to the order-1 statistics.This is obvious in figure 1 where higher order statistics all perform similarly on the left. In figure 2 left trigram performs similarly as well.

Thus this experiment does not provide convincing support of the hypothesis, and can be indicative of the model still using order-k solutions on an order-1 sequence. Since this is the main experiment of the paper, I recommend rejection.

---

> ### Author Response · Authors · 2025-06-03
>
> Thank you for your time, careful reading and feedback. We appreciate your recognition of our paper's contribution in investigating, clearly experimenting with, and theoretically supporting a novel hypothesis.
>
> **[Link to PDF](https://anonymous.4open.science/r/colm25rbtl-C802/bayesian_occam_icl_rebuttal.pdf)** with additional results.
>
>
> **“Results seem to not clearly support the claim that on order-1 chains the model resorts to using the simpler hypothesis.”**
>
> We appreciate your careful reading, and we recognize the importance of clearly demonstrating the results. We maintain that our argument for the Occam's razor-like inductive bias is strongly supported by a comprehensive look at our experimental evidence. Let us clarify our interpretation with further detail.
>
> **1. Figure 1 (Order-1 vs. Order-3 Comparison):** For order-1 contexts, we do observe a clear separation in Figure 1, where KL (bigram ‖ model) ≈ 0.004 is an order of magnitude smaller than KL (tetragram ‖ model) ≈ 0.05. We agree the figure is somewhat cluttered, and we will add these numerical values to the figure caption to emphasize this significant gap, which indicates the model's stronger alignment with the simpler bigram statistics.
>
> **2. Effect of Context Length (Figure 7 in App. A.2):** Your point about Equation (5) and statistics becoming equal for long sequences is crucial. Our experimental design accounts for this prerequisite. Fig. 7 in App. A.2 clearly illustrates this: on order-1 sequences, KL(bigram ‖ model) stays essentially flat and low, indicating a consistent fit to the simplest statistic. In contrast, KL(tetragram ‖ model) is large for short contexts (where higher and lower-order statistics are less similar), and steadily drops as the context grows (as expected by Eq. 5). The clear gap for short contexts validates our hypothesis that the model prioritizes the simpler explanation.
>
> **3. Figure 2 (Order-1 vs. Order-2 Comparison):** You rightly note that the gap is smaller in Fig. 2 when comparing order-1 and order-2 statistics. This is natural, as the complexity difference between these two categories is minimal and from the point above we cannot expect a large gap for large sequence lengths. We acknowledge that this might not be our most indicative comparison. We are running an additional experiment with shorter-length evaluation which we will report and use to replace this figure.
>
> **4. Failure to Generalize from Complex-Only Training (Appendix B):** Our finding in Appendix B is central to our argument. Here, we found that a model trained exclusively on complex (order-3) sequences **fails to infer the correct statistics of an order-1 sequence** when presented with one (Fig. 5, left, in App. B). If the model in Fig. 1 (trained on both order-1 and order-3 sequences) was also merely applying its most complex learned hypothesis, it would behave similarly as Fig. 5, whereas we observe in Fig. 1 that it has lowest KL divergence with the bigram statistics on order-1 sequences. These observations demonstrate a **selective preference** rather than a passive fit when training on a mixture of orders, strongly supporting our Occam's razor hypothesis.
>
> **5. Occam's-Razor Inductive Bias Consistent Across Testbeds:** The same implicit Occam's razor-like bias clearly reappears in our second suite of experiments with linear regressors. In Figures 3a (left) and 3b (left) in the paper, when the test-context length $T$ is such that $d>T>d/2$ (meaning both hypotheses perfectly fit the context data), the transformer's predictions align far more closely with the "simpler" $d/2$-dimensional least-squares solution.
>
> Across both testbeds, the empirical evidence and our Bayesian analysis consistently point to the same behavior: the transformer balances fit against complexity and selects the simplest adequate explanation.
>
> To summarize, based on your feedback, we will make the following specific revisions to the manuscript:  We will **add numerical values** for KL(bigram ‖ model) and KL(tetragram ‖ model) to the caption of Figure 1 to highlight the clear separation on order-1 contexts.  We will **further emphasize the nuance of context length** in the main text, explaining how Fig. 7 in the paper (reproduced as Fig. 6 in the **attached PDF**) directly accounts for Equation 5 and validates our hypothesis in the relevant short-context regimes.

---

> > ### Author Response · Authors · 2025-06-03
> >
> > **Q1: Fig. 2 legend**
> >
> > Thank you for catching this! The lime-yellow curve is the **uniform** baseline. We do not evaluate wrt to tetragram statistics of the sequence in this plot, because the model in Fig. 2 was trained only on order-1 and order-2 sequences. We will correct the legend in the revision.
> >
> > **Q2: Results for order-$k’$ sequences where $1<k’<k$**
> >
> > This is an interesting question.  As shown in **Figure 7 in Appendix A.2**, we already report results for this ablation: we take the transformer trained only on order-1 and order-3 chains and evaluate it on orders-1, 2, 3, and 4 sequences. For each evaluation, we measure the KL divergence of the model's output compared to uniform/ bigram/ trigram/ tetragram/ pentagram statistics for varying prompt sequence lengths. To further validate these findings, we have repeated this experiment, including confidence intervals to better illustrate the gaps between the curves (see Fig. 6 in the **attached PDF**). The new figure (Fig. 6) in the attached PDF confirms consistency with Figure 7.
> >
> > Consistent with the paper’s main finding, for order-1 sequences, the KL divergence to bigram is the lowest, and for order-3, the KL to tetragram is the lowest. For a concrete quantification serving as a baseline for behaviors under order-2 and order-4, if we fix sequence length = 300, the **relative gap** between the lowest KL values and the second lowest (a larger gap indicates more confident prediction) are **1.618** for **order-1** and **1.074** for **order-3** sequences.
> >
> > For **order-4** sequences, we find that the model does not learn pentagram statistics, but rather returns output based on tetragram statistics. This is intuitive: as the model never saw sequences with underlying order-4 relations during training, it naturally defaults to the highest-order hypothesis it has learned (tetragram) as it is the closest alternative capable of partially fitting the observed sequence.
> >
> > Now, moving to **order-2** sequences, which again the model was never explicitly trained on, the behavior is significantly more nuanced. On the one hand, the KL divergence to bigram is high, making it clear the model does not resort to order-1 chains. On the other hand, across sequence lengths, there is no clear “winner” between the tested statistics: concretely, for sequence length=300, the **relative gap** of KL distances to trigram and tetragram is **0.1073**, significantly smaller than the gaps reported above for order-1 and order-3. Thus, while the model avoids the simplest explanation (bigram), it does not strongly commit to either the more complex explanation (tetragram) or a specific intermediate complexity it has not been directly trained on (trigram).
> >
> > We hope that these clarifications help address your concerns and that you would consider increasing your score.

---

> > ### Author Response · Authors · 2025-06-04
> >
> > **Update for Fig. 2 (order-1 vs order-2 comparison)**
> >
> > In [this figure](https://anonymous.4open.science/r/colm25rbtl-C802/order12_150.png), we show results for a transformer trained on mixture of order 1 and order 2 chains with sequence length 200, when evaluated on sequence length 150 as training progresses. Here, we see a clearer gap between the curves for bigram and trigram in the left figure, with the model predictions most closely match with bigram for order-1 (left) and trigram for order-2 sequences (right), respectively, as expected. We will replace Fig. 2 in the paper with this figure.
> >
> > We also evaluated this model on different context lengths at the end of training, shown in [this figure](https://anonymous.4open.science/r/colm25rbtl-C802/order12_context_lengths.png). We find that the gap between bigram and trigram on order-1 sequences increases as context length becomes smaller. This is consistent with the observations in Fig. 7 (model trained on order-1 and order-3 chains).

---

> > ### Comment · Reviewer_neZS · 2025-06-05
> > **Response to authors**
> >
> > I am convinced by the authors response and their results in the appendix. I suggest however that the authors incorporate and reference these results in the main text, to make the conclusion clearer.
> >
> > I believe the results in the paper, and the theoretical analyses, are interesting and important in and of themselves to the COLM audience. I will raise my score to reflect this.

---

> > ### Author Response · Authors · 2025-06-05
> >
> > Thank you very much for your follow-up and for increasing your score! We are grateful for your positive assessment and valuable suggestions.
> >
> > We will absolutely incorporate and reference the new results in the main text as you suggested, which we agree will make the conclusions clearer.
> >
> > Thank you again for your time and feedback.

---

### Official Review · Reviewer_peHZ · 2025-05-13

**Rating:** 7
**Confidence:** 3
**Ethics Flag:** 1

**Summary:**

While most prior work on in-context learning (ICL) has focused on tasks with fixed complexity, this paper shifts attention to scenarios involving varying levels of task complexity. The authors investigate how transformers handle hierarchical task structures, where more complex categories can fully represent patterns generated by simpler ones. To explore this, the authors outline two illustrative testbeds—Markov chains and linear regression—and demonstrate that, with sufficient training, transformers can distinguish between sequences of varying complexity and apply them appropriately. Additionally, they offer an explanation for the selection mechanism, grounding it in the principles of Occam’s Razor. However, the validation of this theory is somewhat limited, with a brief discussion of a probabilistic grammar setting that lacks in-depth analysis. This raises questions about how the simplified example relates to real-world applications.

**Questions To Authors:**

Why did you select GPT2? Would the behavior change with different models or different architectures?

**Reasons To Accept:**

This is a highly theoretical paper that addresses an important topic, with two testbeds carefully described to explore the issue at hand.

**Reasons To Reject:**

It is not clear that the results fully reflect real-world environments, which is the primary motivation of this paper. The experiments themselves are not thoroughly described, and the use of four non-terminal and tree terminal systems in a probabilistic context-free grammar does not provide a convincing approximation of language complexity. As a result, the findings may not generalize well to more realistic, real-world settings.

---

> ### Author Response · Authors · 2025-06-03
>
> Thank you for the feedback. We are encouraged that you view our work as a theoretically strong contribution to an important topic and that you appreciate our experimental design. We address specific comments below.
>
> **[Link to PDF](https://anonymous.4open.science/r/colm25rbtl-C802/bayesian_occam_icl_rebuttal.pdf)** with additional results.
>
> **“It is not clear that the results fully reflect real-world environments, which is the primary motivation of this paper.” “Findings may not generalize to more realistic, real-world settings.”**
>
> As stated in the introduction, our goal is **not** to replicate every facet of real-world languages, but to isolate and study one dimension that existing synthetic setups overlook: **hierarchical task complexity**. As you rightly point out in your summary of our paper, prior work has largely examined fixed-complexity tasks; in contrast, we carefully construct two (Markov-chain and linear-regression) testbeds that deliberately span multiple, nested hypothesis categories. This setup systematically shows whether transformers lean towards simpler or more complex explanations in a clean, controlled setting, uncovering our finding of an Occam's razor-like inductive bias. We believe this controlled approach is essential for rigorously establishing the existence of this inductive bias.
>
> While our primary focus is on uncovering and explaining an underlying mechanism of in-context learning in the most transparent terms, we acknowledge the importance of demonstrating the relevance of our findings to more realistic settings. To that end, motivated by your questions, we have conducted additional experiments on pretrained LLMs (please see response 3 in the **attached PDF**, where we include results for a pre-trained GPT-4 model). These new results provide evidence that similar inductive biases can also be observed in larger, pre-trained models.
>
> **“The experiments themselves are not thoroughly described”**
>
> Given your acknowledgment elsewhere of our "two testbeds carefully described" (referring to our main Markov Chain and Linear Regression testbeds, which are indeed detailed in the main body), we suspect this concern primarily refers to the probabilistic context-free grammar (PCFG) experiment. You are correct that, in optimizing for space in the main paper, we were brief in describing this particular experiment, deferring additional details to Appendices A.4 and C. To address this, we will enhance the description of the PCFG experiment in the main body of the revised manuscript to ensure its design is clearer. If you have specific questions about the setup after reviewing these appendices, please let us know; we are happy to provide further explanations.
>
> **Q: Why select GPT2? Effect of changing model architecture?**
>
> We chose GPT-2 style transformers as our primary architecture due to their widespread adoption in prior studies on ICL with linear-regression and Markov-chain tasks (e.g., Garg et al., 2022; Park et al, 2024, Edelman et al. 2024, Akyürek et al., 2023). Adopting this commonly used architecture allows for direct comparison and builds upon existing insights within this research line.
>
> To assess the impact of different model architectures and scales, we have conducted additional experiments on LSTM architectures and a model scale ablation. Please see response 1 and 2 in the **attached PDF** for details. We find that LSTMs exhibit a slightly weaker form of Occam’s razor-like inductive bias, as they require significantly more capacity to consistently select the correct underlying Markov chain order from the input sequence at inference time compared to transformers. Regarding model scale, we find that larger models also exhibit the Occam’s razor-like inductive bias, and converge faster as the number of parameters is increased. We will add these experiments and discussions in the revision.
>
> We hope that these responses help address your concern and that you will consider increasing your score.

---

> > ### Comment · Reviewer_peHZ · 2025-06-06
> >
> > Thank you for your response, I modified the score.

---

> ### Author Response · Authors · 2025-06-07
>
> Thank you very much for your follow-up and for increasing your score! We appreciate your positive assessment and thank you again for your time and helpful feedback.

---

### Official Review · Reviewer_BYfu · 2025-05-13

**Rating:** 8
**Confidence:** 3
**Ethics Flag:** 1

**Summary:**

The paper studies how autoregressive transformers trained for in-context learning (ICL) handle tasks that have a hierarchical structure, where simpler instances can be perfectly represented by more complex ones. The authors consider 3 synthetic tasks: Markov chains with order k (increasing), dense or sparse linear regression, and probabilistic context-free grammars (PCFGs). Through experiments, they empirically show that models trained on data drawn from a mixture of simple and complex tasks (each separately shown during training) select the simplest hypothesis that fits the context during inference. They also provide some Bayesian based arguments, arguing that this observation is a consequence of marginal-likelihood trade-offs between data fit and model complexity.

**Questions To Authors:**

What do you see as the key implications of your findings for real-world downstream tasks?

**Reasons To Accept:**

- Investigating whether transformers exhibit an implicit “Occam’s razor” during ICL is an interesting research problem that was relatively unexplored from previous studies; this work sheds more lights in models’ inductive biases.

- The authors present strong empirical evidence supporting the hypothesis that LLMs trained for ICL on a mixture of simple and complex task instances (where the latter are able to fully represent the former) select the simplest one during inference. In fact, I think that the experiment in Appendix B, where the authors show that a model trained only on a complex instance does not infer the statistic of a simpler one is particularly interesting, and thus does not generalize well.

- The Bayesian arguments that the authors present to support their empirical observations seem convincing, and rely on reasonable assumptions that agree with their experimental setting.

**Reasons To Reject:**

- Although the authors present some ablations on model scale in Appendix D (especially considering models with fewer parameters), I think that exploring how the main observations change at different (especially higher) scales would be an insightful addition.

- The paper never varies the proportion or ordering of simple vs. complex instances during training. Thus, it is unclear whether the simplicity bias depends on the specific mixture or could be amplified/mitigated by curriculum-style scheduling as the amount of training data grows, or the complexity of the instances is slowly increased.

---

> ### Author Response · Authors · 2025-06-03
>
> Thank you for your detailed review and positive comments on our paper. We are encouraged that you recognize the interesting nature of the research problem we study, and that our results shed light on the model's inductive biases. We also appreciate your acknowledgment of the strong empirical evidence supporting our findings and the good theoretical exposition.
>
> Your comments have been very useful and motivating for additional experiments, which we discuss below and will add to the revision.
>
> **[Link to PDF](https://anonymous.4open.science/r/colm25rbtl-C802/bayesian_occam_icl_rebuttal.pdf)** with additional results.
>
> **Effect of model scale**
>
> Please see response 2 in the **attached PDF**, where we include results for larger transformer models by increasing the number of layers, attention heads, and the embedding dimension. We find that larger models also exhibit the Occam’s razor-like inductive bias, and converge faster as the number of parameters is increased.
>
> **Varying proportion of simple and complex categories during training**
>
> Thank you for this great suggestion. We train transformers while systematically varying the fraction of order-1 and order-3 sequences in each mini-batch (order-1 fraction $\in [0.1, 0.9]$). For each setting, we evaluate at inference time: (i) $\text{KL}(\text{bigram} \|\| \text{model})$ and (ii) $\text{KL}(\text{tetragram} \|\| \text{model})$ on both order-1 and order-3 test sequences. The results are shown as heatmaps in Fig. 4 of the **attached PDF**.
>
> We find that the transformer reliably learns bigram statistics for order-1 sequences, even when order-1 examples make up a small fraction of the training mix. In contrast, learning tetragram statistics for order-3 sequences becomes increasingly difficult as the fraction of order-1 sequences rises—under such skew, the model tends to default to bigram-like behaviour. Moreover, the number of training steps required to acquire tetragram statistics increases as the training distribution becomes more imbalanced.
>
> Increasing the batch size helps offset this effect: larger batches ensure that more order-3 sequences are seen per step in absolute terms, which raises the threshold at which the model can still learn the correct higher-order statistics.
>
> **Effect of curriculum-style scheduling**
>
> That’s another good suggestion. We actually used curriculum learning in the linear regression setting in the paper, following previous linear-regression ICL studies (e.g., Garg et al., 2022 and follow ups). To test its impact, we reran the experiment without any curriculum (Fig. 5, left, in the **attached PDF**). In that setting, the transformer eventually converges to the $d/2$-dimensional least-squares solution, but it takes substantially longer than with curriculum scheduling (Fig. 5, right). We will include this comparison and discussion in the revised version.
>
>
> **Q1: Key implications for real-world downstream tasks**
>
> Motivated by your questions, we have added experiments for a pre-trained GPT-4 model testing whether similar Occam's razor-like inductive biases are observed in larger, pre-trained models (please see response 3 in the **attached PDF**). Our findings suggest that the pretrained model also exhibits a strong preference to the simplest possible explanation among those that explain the context.
>
> This aligns with and provides a fundamental understanding for observations in other relevant empirical work on pretrained LLMs, such as studies on ambiguous examples (Si et al., 2023; Bartsch et al., 2023). Specifically, Si et al. (2023) found that LLMs consistently favor one hypothesis over another when presented with ambiguous sentence-label pairs. Our investigation, using carefully crafted synthetic tasks, allows us to systematically vary task complexity and directly attribute inductive biases to this complexity. This can be seen as a first-principles approach to explaining such empirically observed behaviors in larger models.
>
> While our primary aim is to uncover and explain an underlying ICL mechanism in the most transparent terms, demonstrating broader relevance is important. The new results on pretrained LLMs support the generality of our findings and help bridge our mechanistic insights to these more empirical works.
>
> Overall, we can interpret our findings as suggesting the importance of carefully selecting in-context examples that truly reflect the desired task complexity. If simpler patterns are present, models might favor those in their predictions
>
>
> Thanks a lot for the time and effort to review our work. We appreciate your positive endorsement and support.
>
> **References:**
>
> Si et al., “Measuring Inductive Biases of In-Context Learning with Underspecified Demonstrations”, ACL 2023.
>
> Bartsch et al. “Self-Consistency of Large Language Models under Ambiguity”, BlackboxNLP Workshop, ACL 2023.

---

> > ### Comment · Reviewer_BYfu · 2025-06-07
> >
> > Thank you for your response and the additional experimental results. You’ve addressed all my concerns thoroughly, so I’ll be increasing my score.

---

> ### Author Response · Authors · 2025-06-07
>
> Thank you very much for your follow-up and for increasing your score! We appreciate your positive endorsement and support. Thank you again for your great suggestions that helped improve our work!

---

### Official Review · Reviewer_Qdck · 2025-05-14

**Rating:** 7
**Confidence:** 4
**Ethics Flag:** 1

**Summary:**

The authors formulate hierarchical tasks where higher complexity tasks can strictly represent lower complexity tasks (example: k-order markov chains), they then examine whether transformer architectures do ICL to learn orders of task complexity, and choose the simplest possible interpretation of task given in the prompt at inference time. The authors empirically assert that transformer models can discover different complexity levels for hierarchical tasks, and will prefer the simplest possible explanation. In practice the evaluation is accomplished by measuring the kl-divergence between model distribution and different order statistics in order to measure whether models can switch to different complexity levels as a function of input complexity. This work builds on related approaches which focus on either specific tasks or finite sets of tasks, to examine tasks which are explicitly hierarchical -- specifically predicting continuations of k-order markov chains. The authors present a Bayesian interpretation of results, showing that the transformer architecture can learn to select the simplest hypothesis when trained on tasks that enable hierarchical factorization.

The paper is clearly written with a good theoretical exposition, but most of the content is focused upon the problem and theoretical framing rather than upon anything specific to the transformer architecture, or to LLMs in general. The key takeaways of the paper aren't really clear, but the insights are still probably interesting for a subset of the COLM audience interested in bertology-related topics.

**Questions To Authors:**

for the COLM audience, suggest moving more of the mathematical exposition to the appendix and adding more analysis of the transformer architecture itself

**Reasons To Accept:**

- clear exposition and theoretical motivation of synthetic tasks that measure the transformer architecture's ability to learn hierarchical task representations
- good experimental setup that should be straightforward to replicate and extend to new tasks

**Reasons To Reject:**

- low applicability in the current presentation to real world LLM evaluations, no examination of how results may generalize to i.e. foundation models
- no examination of other possible architectures such as LSTMs -- is this a unique characteristic of transformer models? Do they need to be a certain size / dimension etc as a function of task complexity, etc ...
- the takeaway for readers aren't really clear beyond "here's a cool observation about transformer models in this specific setting"

---

> ### Author Response · Authors · 2025-06-03
>
> Thank you for the detailed comments and your overall positive feedback. We are encouraged that you found the paper well-written with theoretically motivated synthetic tasks for investigating hierarchical task learning in transformers, the insights interesting, and the experimental setup replicable and extensible to new tasks.
>
> Your comments have been very useful and motivating for additional experiments, which we discuss below and will add to the revision.
>
> **[Link to PDF](https://anonymous.4open.science/r/colm25rbtl-C802/bayesian_occam_icl_rebuttal.pdf)** with additional results.
>
> **Other Architectures such as LSTMs**
>
> Thank you for the suggestion. Please see response 1 in the **attached PDF**, where we include results for LSTMs with different number of layers and embedding dimensions. We find that LSTMs exhibit a weaker form of Occam’s razor-like inductive bias, as they require significantly more capacity to consistently select the correct underlying Markov chain order from the input sequence at inference time compared to transformers.
>
> You rightly point out that our theoretical framing is not tied to any specific architecture. In that sense, the fact that LSTMs eventually, with sufficient scaling, can exhibit similar behavior is consistent with our analysis. However, the observed difference in efficiency and the pronounced nature of the bias in transformers confirms a stronger architectural inductive bias in transformers for this type of hierarchical task learning.
>
> **Effect of Model Size**
>
> Please see response 2 in the **attached PDF**, where we include results for larger transformer models by increasing the number of layers, attention heads, and the embedding dimension. We find that larger models also exhibit the Occam’s razor-like inductive bias, and converge faster as the number of parameters is increased.
>
> Furthermore, these ablations confirm that a smaller transformer with 6-layers, 6-heads fails to learn tetragram statistics for order-3 sequences, directly supporting your intuition that a minimum model size is required to handle a given task complexity. The LSTM experiments support a similar conclusion. Thank you for suggesting this experiment, it will be a valuable addition to the revised version.
>
>
> **Examination of how results may generalize to real-world LLMs**
>
> Thank you for the suggestion. Please see response 3 in the **attached PDF**, where we include results for a pre-trained GPT-4 model testing whether similar Occam's razor-like inductive biases are observed in larger, pre-trained models. Our results suggest that the pretrained model also exhibits a strong preference to the simplest possible explanation among those that explain the context.
>
> Overall, as you note, the ideas and insights in the paper can extend to other tasks that exhibit a hierarchy of complexity levels, beyond the settings we consider. Several recent works leverage synthetic setups (e.g., linear regression, Markov chains, Boolean functions), to study various aspects of ICL in transformers, such as which function classes can be learned in-context, what algorithm is used for ICL of linear regression, the transient nature of ICL, phase transition based on diversity of training tasks, and so on (as discussed in Section 2.2 in the paper). These studies, though grounded in synthetic settings, provide important insights into the ICL behaviour of small-scale transformers, and help inform our understanding of ICL in larger, real-world models. Our work follows a similar philosophy, focusing on ICL in settings with hierarchical task complexity, where we uncover an Occam's razor-like inductive bias.
>
> While our primary aim is to uncover and explain an underlying mechanism of ICL in the most transparent terms, we acknowledge that demonstrating broader relevance is important. To that end, the new results on pretrained LLMs, which provide evidence that similar inductive biases can also be observed in larger, pre-trained models, will be included in the revision to support the generality of our findings.
>
> **Moving Mathematical Exposition to Appendix**
>
> We appreciate this suggestion. We agree on the importance of maintaining clarity and accessibility for the COLM audience while still providing the necessary transparency into our findings. To balance things, we will move Section 3.2.3 (Analysis for the Linear Regression Setup) to the Appendix. In its place, we will expand the analysis and discussions related to the new experiments prompted by your questions—specifically, our evaluations on pretrained models and comparisons to LSTMs.
>
>
> Thanks a lot for the time and effort to review our work. We appreciate your positive endorsement and support.

---

> > ### Comment · Reviewer_Qdck · 2025-06-09
> >
> > thank you for the thorough response and the additional information, I'm bumping my score to 7

---

> > > ### Author Response · Authors · 2025-06-09
> > >
> > > Thank you very much for your follow-up and for increasing your score! We are grateful for your positive assessment and valuable suggestions, they really helped improve our work. We will add the additional experiments to the revised version.

---

### Author Response · Authors · 2025-06-03
**General Response to Reviewers**

We sincerely thank all the reviewers for their time and effort in reviewing our work, as well as for their thoughtful feedback and helpful suggestions. We have carefully considered all comments and have conducted additional experiments and analyses to address the points raised.

We've gathered all the additional experiments and a summary of their main results in the **attached PDF** below. We address the other comments in the individual responses to each reviewer, with pointed references to this attached PDF as needed.


### **[Link to PDF](https://anonymous.4open.science/r/colm25rbtl-C802/bayesian_occam_icl_rebuttal.pdf)**


Here are the main results from the additional experiments:

**1: Experiments with LSTMs (Revs Qdck, peHZ)**

We include results for LSTMs with different layers and embedding dimensions (Fig. 1). Our results show that LSTMs exhibit a slightly weaker form of Occam’s razor-like inductive bias, as they require significantly more capacity to consistently select the correct underlying Markov chain order from the input sequence at inference time compared to transformers.

**2: Effect of model scale (Revs Qdck, BYfu, peHZ)**

We include results for larger transformer models by increasing the number of layers, attention heads, and the embedding dimension (Fig. 2). We find that larger models also exhibit the Occam’s razor-like inductive bias, and converge faster as the number of parameters is increased.

**3: Pre-trained LLMs (Revs Qdck, BYfu, peHZ)**

We include results for a pre-trained GPT-4 model testing whether similar Occam's razor-like inductive biases are observed in larger, pre-trained models (Fig. 3). Our results suggest that the pretrained model also exhibits a strong preference to the simplest possible explanation among those that explain the context.

**4. Effect of varying the training mixture proportions and curriculum training (Rev BYfu)**

In Fig. 4, we train transformers on a mixture of order-1 and order-3 sequences for increasing fractions of order-1 sequences in each mini-batch. Our results show that even when this fraction is very low, the model learns bigram statistics for order-1 sequences. Increasing this fraction makes the learning of tetragram statistics for order-3 sequences increasingly difficult, but increasing the batch size helps offset this effect.

In Fig. 5, we compare the effect of training transformers without/with curriculum learning in the linear regression setting and observe that curriculum speeds up convergence.

---

### Decision · Program_Chairs · 2025-07-08

**Decision:**

Accept

**Comment:**

This paper investigates an important and novel hypothesis: Transformers' ICL inherently favor simpler hypotheses when given tasks structured hierarchically by complexity. The authors provide clear theoretical framing based on Bayesian principles and experimentally demonstrate this "Occam's razor"-like inductive bias using well-designed synthetic testbeds (Markov chains, linear regression, and probabilistic grammars).

All reviewers agree on the clarity, theoretical depth, and novelty of the hypothesis, along with the strength of the controlled experiments. Initial concerns centered around limited experiments regarding real-world applicability, model scale generalization, and potential ambiguities in interpreting Markov chain results. Authors' detailed rebuttals successfully addressed most major concerns through additional experiments: verifying the inductive bias in larger models (GPT-4), exploring model architectures (LSTMs), clarifying the Markov chain experiment interpretations, and examining curriculum and task mixture effects.

That said, I am surprised to see that the paper currently lacks a discussion on structural risk minimization (SRM), an essential theoretical context for understanding inductive biases and model complexity trade-offs. The authors must explicitly address this relationship in their revision to fully situate their findings within the broader theoretical literature.

Overall, the paper effectively establishes ICL's inductive biases toward simpler explanations in hierarchical contexts, with empirical evidence. The improvements promised in the authors’ rebuttals, along with a needed discussion on SRM, would enhance the paper's clarity, applicability, and generalizability.